# The Firefighter problem with dynamic defence costs

Ethan Hunter ![ORCID]*, Jessica Enright

School of Computing Science, University of Glasgow, Glasgow, United Kingdom

* e.hunter.2@research.gla.ac.uk

## Abstract

The Firefighter Problem is a single-player game modelling the spread of a contagion (e.g. rumours, diseases) on a graph. The player's objective is to defend vertices to protect at least a given number. This problem is computationally hard, but it can be solved efficiently on certain restricted classes of graph, such as complete graphs (in constant time) and graphs with path length at most $\ell - 1$ (in $\mathcal{O}((n + m)n^{\ell - 2})$-time). We define The Cost Function Firefighter Problem, the first variant of the Firefighter problem that introduces vertex defence costs depending on time and game state. We show the Cost Function problem is computationally hard even for classes of trees on which the classic problem is tractable, but tractable on some very restricted graph classes (complete graphs, graphs of fixed bounded path length and trees under certain conditions). By expressing our variant in monadic second-order logic, we prove it is fixed-parameter tractable with respect to treewidth, budget, and maximum time step.

To complement theoretical findings, we undertake empirical investigation to compare performance of cost, threat and degree-based heuristics under various cost functions. We find that the relative effectiveness of these heuristics depends heavily on graph structure, with degree-based heuristics generally performing worse than state-based strategies. We show how these heuristics play out on both random and real-world interaction graphs.

## 1 Introduction

The Firefighter Problem (FIRE) is a single-player, discrete-time game proposed by Hartnell [1] to model the spread of fire or other contagion in a graph. A FIRE instance consists of: a target number of vertices to save, a graph, and a root vertex that serves as the initially burning vertex. Vertices inhabit one of three states at any given time: *open, burning* or *defended* (formalised in Definition 1). The objective is typically to maximise the number of vertices saved from burning. Many variants of FIRE have been studied [2].

We describe a new variant called The Cost Function Firefighter Problem (which we simply call COST-FIRE). The key feature of COST-FIRE is the use of *cost functions*

**Data availability statement:** The contact graph data used for simulations are available under a Creative Commons Attribution-ShareAlike licence: sleepy lizard data at https://doi.org/10.5061/dryad.jk87h and raccoon data at https://doi.org/10.5061/dryad.gr40r. The code used to perform simulations is publicly available at https://github.com/Ethan-CS/CostFnFire. Full simulation results, including random generation seeds and strategies for verification, are publicly available at https://doi.org/10.6084/m9.figshare.30053002.v1.

**Funding:** Enright was partially supported by EPSRC grants: EP/T004878/1 and EP/V032305/1.

**Competing interests:** The authors have declared that no competing interests exist.

to assign defence costs to each vertex that (importantly!) can depend on time and the state of burning in the graph, thus introducing a dynamical aspect. The firefighter is assigned a budget to spend each turn on defending vertices, and may only defend a set of vertices whose costs sum to at most the budget in that turn.

Formally, a decision-problem instance of Cost-Fire consists of a *graph* $G = (V, E)$ (which we formally define in Sect 3.1) with a vertex $r \in V$ designated the root, a computable cost-function $c$ that assigns a positive integer cost to each vertex (given a game state and time step), a budget $b \in \mathbb{N}$ and a target $k \in \mathbb{N}$. The question is whether there exists a sequence of defences that obeys the budget at each time step while saving at least $k$ vertices by the end of the game. An optimisation version would instead ask for the maximum number of vertices that could be saved with a given budget. For computational complexity purposes, the graph size, budget, and target are considered part of the instance size, with an oracle assumed for $c$.

## 1.1 Motivation

We have designed Cost-Fire to model population heterogeneity, such as variations in vaccine hesitancy. A recent survey by Pourrazavi et al. analysed 91 articles reporting COVID-19 vaccine uptake data, across several countries and programs, revealing an average hesitancy rate of 29.72% [3]. In Sect 5, we design cost functions with this rate to model individual inclination towards being defended for heuristic experiments.

## 1.2 Contribution and structure

We present related work on Fire in Sect 2 as foundation for the formal definition of the new Firefighter variant in Sect 3.3. Cost-Fire generalises Fire by introducing variable defence costs using *cost functions* that depend on time and the game state, and by imposing a per-turn defence budget. We distinguish between *temporal* and *static* cost functions, which do and do not depend upon time respectively, and explore examples of each in Sect 3.4. These cost functions can be used to model, for example, vaccine hesitancy, inclination towards personal protection, and individuals influencing those around them. In particular, we can design cost functions that model these behaviours evolving over time.

As a generalisation of Fire, Cost-Fire is clearly NP-hard (Theorem 3). Following common practice in Fire variant research [2], in Sect 4 we explore graph classes on which Cost-Fire can be efficiently solved to better understand the hardness of the problem. In Sect 4.1, we provide a reduction from a SAT variant to show that Cost-Fire remains hard on a highly symmetric sub-class of trees where Fire is tractable. We briefly prove that Cost-Fire on complete graphs admits an algorithm that is polynomial in graph size and explain why Fire admits a linear-time algorithm by similar reasoning in Sect 4.2. Then, in Sect 4.3, we prove that Cost-Fire is, like Fire, tractable on graphs with fixed maximum path length, yielding an immediate parametrised result for Cost-Fire.

With appropriate cost functions, we can re-couple the budget-degree relation, allowing extensions of a well-known strategy for Fire on trees of bounded degree [4]. This renders Cost-Fire tractable on such trees with these cost functions, as shown in Sect 4.4.

We summarise tractability and hardness results for Cost-Fire (and Fire for comparison) on the considered graph classes in Table 1.

In Sect 4.5, we use a result due to Courcelle [6] to show that Cost-Fire admits a fixed-parameter tractable algorithm by the sum of: the treewidth of the input graph, the budget and the maximum time step.

We present experimental data in Sect 5, the results of a custom Python script to simulate Cost-Fire on various cost functions and graphs to compare the performance of different heuristic defence strategies. These results show heuristic performance varying based on graph structure, particularly density. Notably, degree-based defence strategies frequently save the fewest median number of vertices, while threat- and cost-based heuristics perform similarly.

## 2 Related work

We use standard computational complexity notions throughout, including NP-hardness, associated reduction proof techniques and the parameterised complexity classes fixed-parameter tractable (FPT) and slicewise polynomial (XP). For the unfamiliar interested reader, we recommend the introductory text by Cormen et al. [7], or for an introduction to parametrised algorithmics the text by Cygan et al. [8]. A *graph* is a pair consisting of a set of *vertices $V$* and a binary operation on them $E$ that defines the *edges*; we define required graph theoretic concepts in Sect 3.1.

### 2.1 The Firefighter problem

We recommend the comprehensive survey of the classic Firefighter problem by Finbow and MacGillivray [2]. As input for Fire, we require: a rooted graph $(G = (V, E), r \in V)$ and an integer $k$. Throughout the game, each vertex inhabits one of the following game states at each turn.

**Definition 1** (Fire problem states). All vertices except the root are initially *open*, meaning neither burning nor defended. Open vertices adjacent to burning vertices catch fire to become *burning* in the subsequent turn. Open vertices can be selected by the firefighter (player) to become *defended*, meaning they cannot catch nor transmit fire. Once burning or defended, a vertex remains so for the rest of the game.

A *strategy* for an instance of Fire is a sequence of vertices selected for defence, indexed by time of selection. An instance of Fire ends when fire is *contained*, i.e. when no open vertex is adjacent to a burning vertex. At this point, we declare all unburned vertices *saved*.

---

Firefighter (Fire) [2]

**Instance:** A rooted graph $(G = (V, E), r)$ and an integer $k \geq 1$.

**Question:** Is there a finite sequence of vertices $d_1, d_2, \ldots, d_t$ for some positive integer $t \leq |V|$ such that if fire breaks out at $r$ then:

- $d_i$ is neither burning nor defended at time $i$,
- at time $t$, fire is contained and at least $k$ vertices saved by defending each $d_i$ at time $i$?

---

**Table 1. Comparison of tractability and hardness results for FIRE and COST-FIRE for graphs on $n$ vertices and $m$ edges, maximum path length $\ell$ and budget $b$.** PATHCONTAINABLE is defined in Sect 4.4.

| Graph class | FIRE | COST-FIRE |
|---|---|---|
| *Sea Fan* | $O(n)$ (§ 4.1) | NP-hard (§ 4.1) |
| *Complete* | $O(1)$ | $\mathcal{O}(n \ln n)$ (§ 4.2) |
| $P_\ell$-free | $O(n^\ell)$ [5] | $\mathcal{O}((n + m)n^{b(\ell-2)})$ (§ 4.3) |
| *Tree* | NP-hard; $O(n + m)$ if max. children is 2 [4] | NP-hard; $\mathcal{O}(n + m)$ if PATHCONTAINABLE (§ 4.4) |

Finbow et al. give a reduction from a variant of the Boolean Satisfiability Problem (SAT) to show that FIRE is NP-hard even when restricted to trees of maximum degree three, although an algorithm that is polynomial in tree size exists when the root has degree 2 [4]. We rely on the hardness result for reductions and use the tractability result for comparison in Sect 4.4.

**Theorem 1.** FIRE *is NP-hard even when restricted to trees of maximum degree three [4].*

**2.1.1 Tractability.** There are several graph classes on which FIRE is known to be resolvable in time polynomial in graph size. For example, FIRE can be decided in constant time when the the root vertex is adjacent to every other vertex in the input graph (e.g. complete graphs). Intuitively, this is because we can save only one vertex before every remaining vertex is burned in the first turn, hence the question simplifies to asking whether $k \leq 1$. We prove a result using a similar observation for COST-FIRE on such graphs in Sect 4. Other graph classes on which FIRE is known to be solvable in time polynomial in graph size include: graphs of maximum degree three rooted at a vertex of degree at most 2 [4]; $k$-caterpillars for fixed $k$ [9]; $P_k$-free graphs for fixed $k$ [5]; and several perfect graph subclasses (interval graphs, permutation graphs, split graphs and cographs) [5].

**2.1.2 Approximation.** Various approaches have been considered for FIRE with varying success, with many results restricted to FIRE on rooted trees $(T, r)$. An obvious first attempt on trees is to defend vertices greedily, choosing at each step the vertex $v$ whose defence would save the greatest number of descendants Hartnell and Li show that this greedy algorithm produces a 1/2-approximation of FIRE on trees [10]. Cai et al. [11] use a linear programming (LP) relaxation to produce a $(1 - 1/e)$-approximation algorithm for FIRE on trees. This is known to be tight for LP-based rounding approximations [11].

For general input graphs, Anshelevich et al. show that FIRE is not $n^{(1-\epsilon)}$-approximable for any $\epsilon > 0$ unless $P = NP$ [12].

**2.1.3 Variants.** FIRE can be generalised by permitting more than one vertex to be defended per turn - this variant is called The Budget Firefighter Problem ($b$-FIRE). The number of vertices that can be defended is called the *budget.* Bazgan et al. [9] prove the following extension of Theorem 1 for $b$-FIRE.

**Theorem 2.** $b$-FIRE *with fixed budget $b \in \mathbb{N}$ is NP-hard even for trees of maximum degree $b + 2$, but solvable in time polynomial in graph size on such trees when the root has degree at most $b + 1$ [9].*

In COST-FIRE, we allow cost functions to depend on time, thereby introducing *temporality* to the problem by changing the problem state *as a function of time.* To our knowledge, the only other attempt at introducing explicit temporality to FIRE is due to Hand et al. [13]. They define The Temporal Firefighter Problem (TEMP-FIRE), which is FIRE played on *temporal graphs*, graphs whose edges only exist at certain times. The temporal aspect of TEMP-FIRE introduces complexity. For example, FIRE is straightforwardly tractable on complete graphs [4], but TEMP-FIRE is NP-Hard on temporal graphs when the underlying graph is complete. This is also the case for trees and graphs of maximum degree three (or $b + 2$ for budget $b \in \mathbb{N}$) when the root does not have maximum degree.

The Weighted Firefighter Problem (WEIGHT-FIRE) generalises FIRE by assigning to each vertex a static *weight* value and the aim of the defender is to maximise the weights of the saved vertices - see work by Duffy and MacGillivray [14]. While both WEIGHT-FIRE and COST-FIRE introduce values associated with vertices, the weights in WEIGHT-FIRE represent values that we aim to maximise in our saved vertices rather than costs to defend. In contrast, in COST-FIRE, we aim to maximise the *number* of vertices saved, with their costs restricting how many we can save. Duffy and MacGillivray show that WEIGHT-FIRE is not only NP-hard for graphs of maximum degree $b + 2$ when fire starts at a vertex of degree at most $b + 1$ where $b$ is the defence budget but it is NP-hard even for binary trees [14]. However, if weights are restricted to positive real numbers, WEIGHT-FIRE is polynomial-time solvable on binary trees [14].

## 3 Theoretical background

Our theoretical work uses tools of algorithmic graph theory: in this section, we briefly outline some of these tools, and then give the formal problem definition. We also provide basic graph theoretic definitions (predominantly following Wilson [15]) for background and to specify our notational choices.

### 3.1 Graph theoretic preliminaries

For our purposes, graphs are always undirected and unweighted, which is for simplicity. We provide a brief discussion of potential generalisations of our results in Sect 7.2.

Let $G = (V, E)$ be a graph. A *path* between $v_i, v_j \in V$ is a sequence of distinct edges $((v_i, w_1), (w_1, w_2) \ldots, (w_{k-1}, w_k), (w_k, v_j))$ that join $v_i$ to $v_j$. The number of edges on the shortest path between two vertices $u, v \in V$ is called the *distance* between $u$ and $v$, denoted $\text{dist}(u, v)$ (if no such path exists, $\text{dist}(u, v) = \infty$). For a vertex $v$ in a graph, the *neighbourhood* of $v$, denoted $N(v)$, is the set of vertices that share an edge with $v$. A *rooted graph* is a pair $(G = (V, E), r)$ for which a vertex $r \in V$ has been labelled the *root*. For a graph $G = (V, E)$, a *subgraph* $H = (V_H, E_H)$, denoted $H \subseteq G$, is a graph on a vertex-set $V_H \subseteq V$ and edge-set $E_H \subseteq E$ such that $\forall (v_i, v_j) \in E_H, v_i, v_j \in V_H$. An *induced subgraph* of a graph $G = (V, E)$ is a subgraph $H = (V_H, E_H)$ such that $\forall v_i, v_j \in V_H, \ (v_i, v_j) \in E \implies (v_i, v_j) \in E_H$. A graph $G = (V, E)$ is *connected* if, for all vertices $v_i, v_j \in V$, there is a path between $v_i$ and $v_j$. A *tree* is an undirected, connected graph with no cycles. The *level* of a vertex $v$ in a *rooted tree* $(T, r)$ is the minimum distance from $r$ to $v$. In a rooted tree, a vertex $v$ can have *parents* or *children* (or both): a vertex $w$ adjacent to $v$ is said to be a *parent* of $v$ if $\text{dist}(r, w) < \text{dist}(r, v)$ or a *child* of $v$ if $\text{dist}(r, w) > \text{dist}(r, v)$. A *complete graph* is a graph where each vertex is adjacent to every other vertex.

### 3.2 Theoretical methods

To show that COST-FIRE is NP-hard on general input graphs, in Sect 4 we use a *reduction* from FIRE. This involves creating an instance $A$ of COST-FIRE from an arbitrary instance $B$ of FIRE such that there is a polynomial-time transformation between $A$ and $B$ and $A$ is a yes-instance if and only if $B$ is a yes-instance. We use the same general technique in Sect 4.1, this time from an NP-hard variant of the Boolean Satisfiability problem, to show that COST-FIRE remains hard on a class of graph on which FIRE is solvable in time polynomial in graph size. This is a gadget-based reduction.

Proceeding to tractability results, in Sects 4.2 and 4.3 we give straightforward algorithmic proofs that COST-FIRE is tractable on complete graphs and graphs of bounded path length respectively. In Sect 4.4, we define restrictions on cost functions to revive the degree-budget coupling observed by Finbow et al. [4] for $b$-FIRE, leading to a related tractability result for COST-FIRE on trees of specified maximum degree. In Sect 4.5, we express COST-FIRE in extended monadic second-order logic (EMSO). In doing so, we benefit from an extension of a well-known result (Courcelle's theorem) which means COST-FIRE is fixed-parameter tractable by the budget, maximum time step and the treewidth of the input graph.

### 3.3 Problem definition

To formally define COST-FIRE, we first formalise *instances*, *strategies* and *cost functions.* An instance $\mathcal{I}$ of COST-FIRE is a tuple consisting of a rooted graph $(G = (V, E), r)$, a cost function $c$ mapping each vertex to an integer that can depend on time and game state, a budget $b \in \mathbb{N}$ and a target $k \in \mathbb{N}$.

**Definition 2** (Strategy with multiple defences per turn). For an instance of COST-FIRE with budget $b$ on a rooted graph $(G = (V, E), r)$, a strategy is a sequence of sets of vertices $\sigma = (\{d_1^1, d_1^2, \ldots, d_1^{p_1}\}, \{d_2^1, \ldots, d_2^{p_2}\}, \ldots, \{d_t^1, \ldots d_t^{p_t}\})$ for some positive integer $t \leq |V|$. Here, $d_i^j$ denotes the $j$-th vertex defended at time $i$ and $p_i$ is the total number of vertices defended at time $i$.

For convenience, we denote by $\sigma_i$ the set of vertices defended at time $i$.

**E.g. 1.** To illustrate this definition, consider an instance of COST-FIRE on the graph in Fig 1 with root labelled $r$, a cost function that defines the cost to defend all vertices as 1 in all turns, and a budget of 2.

In this instance, fire breaks out at $r$ in the first turn. We then defend $D$ and $H$, after which fire spreads to $A$ and $C$. In turn 2, we defend $B$ and in doing so contain the fire. Hence, the strategy denoted as per Definition 2 is $\sigma = (\{D, H\}, \{B\})$ (and saves 6 vertices).

To allow costs to vary according to the state of the problem, we define the following mapping and give an example to illustrate.

**Definition 3** (State mapping). Let $G$ be a graph on $n$ vertices $v_1, v_2, \dots, v_n$. For an instance $\mathcal{J}$ of COST-FIRE on $G$, given a strategy $\sigma$, let $T$ denote the maximum set of times $[0, |\sigma|] \subset \mathbb{N}$ that the game could last, and define the state mapping

$$\mathcal{S}_{\mathcal{J},\sigma} : T \to \{((v_1, x_1), (v_2, x_2), \dots, (v_n, x_n)) \mid x_i \in \{\texttt{burning}, \texttt{defended}, \texttt{open}\}\}$$

as a function that, for a given instance, strategy played and time, returns the state of the game as a set of vertex-state pairs.

When the instance is unambiguous, we omit the subscript and write $\mathcal{S}_\sigma$, and similarly $\mathcal{S}$ when both instance and strategy are clear.

**E.g. 2.** To illustrate Definition 3, we refer again to the instance of COST-FIRE that we depict in Fig 1, which we label $\mathcal{J}$. Let the graph be denoted $G = (V, E)$. The state mapping for instance $\mathcal{J}$ under the strategy $\sigma = (\{D, H\}, \{B\})$ is as follows.

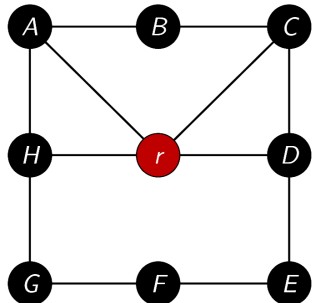

(a) Turn 0: outbreak at $r$

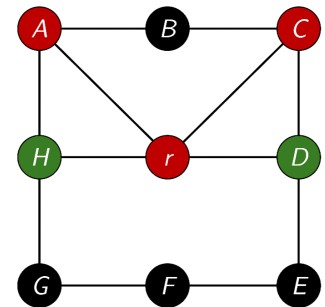

(b) Turn 1: defend $D$ & $H$;
A & C catch fire

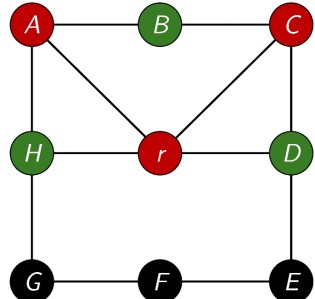

(c) Turn 2: defend $B$; fire
is contained

**Fig 1**. Illustration of a strategy for an instance of **COST-FIRE** on the depicted graph on 9 vertices, rooted at $r$, where each vertex costs 1 to defend and a budget of 2 per turn is available.

$$S_{\mathcal{I},\sigma}(0) = \{(A,\texttt{open}),(B,\texttt{open}),(C,\texttt{open}),(D,\texttt{open}),(E,\texttt{open}),$$
$$(F,\texttt{open}),(G,\texttt{open}),(H,\texttt{open}),(r,\texttt{burning})\}$$
$$S_{\mathcal{I},\sigma}(1) = \{(A,\texttt{burning}),(B,\texttt{open}),(C,\texttt{burning}),(D,\texttt{defended}),(E,\texttt{open}),$$
$$(F,\texttt{open}),(G,\texttt{open}),(H,\texttt{defended}),(r,\texttt{burning})\}$$
$$S_{\mathcal{I},\sigma}(2) = \{(A,\texttt{burning}),(B,\texttt{defended}),(C,\texttt{burning}),(D,\texttt{defended}),(E,\texttt{open}),$$
$$(F,\texttt{open}),(G,\texttt{open}),(H,\texttt{defended}),(r,\texttt{burning})\}$$

For convenience, denote by $\mathfrak{S}_{\mathcal{I}}$ the set of all possible game states, which we illustrate with the following example.

**E.g. 3.** Consider an instance $\mathcal{I}$ of COST-FIRE on a path on two vertices, $u$ and $v$, rooted on $u$, with budget and costs such that we can implement a strategy $\sigma = (\{v\})$ (that defends $v$ at time 1). The state mapping $S_{\mathcal{I},\sigma}$ at time 1 is $S_{\mathcal{I},\sigma}(1) = \{(u,\texttt{burning}),(v,\texttt{defended})\}$. Note that $\mathfrak{S}_{\mathcal{I}}$ contains elements such as $\{(u,\texttt{burning}),(v,\texttt{defended})\}$ (which applies to this example) and $\{(u,\texttt{burning}),(v,\texttt{burning})\}$ (possible given no defence).

We are now equipped to formally define COST-FIRE.

---

COST FUNCTION FIREFIGHTER (COST-FIRE)

**Instance:**

- A rooted graph $(G = (V,E), r)$,
- a cost function $c : V \times \mathbb{N} \times \mathfrak{S} \to \mathbb{N}$, an oracle mapping each $v \in V$ to an integer cost based on time and game state,
- a budget $b \in \mathbb{N}$, and
- an integer $k \geq 1$.

**Question:** Is there a strategy $\sigma$ (as in Definition 2) such that, if fire breaks out at $r$ and we defend using $\sigma$:

- for each $\sigma_i \in \sigma$:
  - each $d \in \sigma_i$ is open at time $i$–1, and
  - $\sum_{d \in \sigma_i} c(d, i, S(i, \sigma_{i-1})) \leq b$, and
- at time $|\sigma|$, fire is contained and at least $k$ vertices saved?

---

We define the cost function as an oracle because $\mathfrak{S}$ takes, in general, space exponential in the number of vertices to calculate. However, in practice we keep track only of the state of the game being played. This definition requires budgets and costs to be integers for simplicity - rationals could also be used and proofs would transfer subject to a multiplicative factor.

### 3.4 Examples of cost functions

Cost functions can depend on: vertex, problem state and turn. Different combinations of parameters are more appropriate for different contexts, so we explicitly characterise them as follows:

- $c(v)$ - *static,* depends only on the vertex,
- $c(v,t)$ - *purely temporal,* depends on vertex and turn,

- $c(v, \mathcal{S})$ - *purely state-dependent,* depends on vertex and current problem state, and
- $c(v, \mathcal{S}, t)$ - *temporally state-dependent,* depending on vertex, problem state and turn.

Graph state driving changes in costs can model, for instance, individuals in an epidemic model being more easily convinced to seek a vaccination if a close contact is infected. Purely temporal functions can be used to model, for example, the effect of seasonality on a spreading infection. To build intuition, we give some example cost functions: for each, consider an instance of COST-FIRE on a rooted graph $(G = (V, E), r)$ and a budget $b \in \mathbb{N}$.

**E.g. 4** (Randomised individual preference). Let $c : V \to \mathbb{N}$ be $c(v) \mapsto U[1, 5]$, where $U[1,5]$ indicates an integer sampled uniformly at random from the range [1,5].

This function assigns each vertex a uniformly random cost at the start of the problem, which does not change as the game progresses. In particular, this depends on neither time nor problem state, so it is clearly *static*.

For the next example, let $f_{v,t}$ denote the number of burning vertices in the neighbourhood of $v \in V$ at time $t$.

**E.g. 5** (Burning neighbours). Let $c : V \times \mathfrak{S} \to \mathbb{N}$ be

$$c(v, \mathcal{S}(t, \{d_1^1, \dots, d_t^{r_t}\})) = \max\{b - f_{v,t}, 1\}.$$

This function sets the cost to defend any vertex as the budget $b$ minus the number of burning neighbours that vertex has. In an epidemic context, this *purely state-dependent* function could model individuals more easily adopting preventative measures as more contacts become infected.

**E.g. 6** (Distance from closest fire). We define $c : V \times \mathfrak{S} \to \mathbb{N}$ as

$$c(v, \mathcal{S}(t, \{d_1^1, \dots, d_t^{r_t}\})) = \min\{\text{dist}(u, v) \mid (u, \texttt{burning}) \in \mathcal{S}(t, \{d_1^1, \dots, d_t^{r_t}\})\}.$$

That is, the cost to defend each vertex is equal to the shortest distance to the closest burning vertex in the graph. This is another example of a *purely state-dependent* function. Because it calculates the distance from closest fire, this may be a computationally expensive function. Note that under this cost function, individuals only become defensible once fire is within a certain proximity.

**E.g. 7** (Purely temporal example). Let $c : V \times [0, |V|] \to \mathbb{N}$ be a function that assigns costs to vertices based on time as:

$$c(v, t) = \begin{cases} 2 & \text{if } 2 \mid t \\ 1 & \text{otherwise} \end{cases}$$

This function could be used as a basic way of studying defence becoming more difficult seasonally - every second turn, defence is twice as expensive for all vertices. Since this is independent of problem state, this is a *purely temporal* function. We could combine this notion with example 6 to create a function that is both periodic and changes based on game state to define a *temporally state-dependent* cost function.

Various modelling dynamics can be easily represented using cost functions. For example, delay (modelling e.g. a given time required to deploy defence) can be modelled by assigning costs that exceed the available budget until a certain turn or game state has been reached. Further, uncertainty can be modelled using a stochastic function to determine the cost to defend, which can be used to represent incomplete or noisy information about the defensibility of individual vertices.

We can also introduce feedback loops (modelling e.g. adaptive behaviour in a population) using cost functions that respond to the game state in previous turns.

## 4 Theoretical results

To place the hardness of COST-FIRE in the landscape of FIRE and variants, we begin by proving the problem is NP-complete on general input graphs before showing that it remains hard on a class of highly symmetric trees where FIRE is tractable. We then give some simple cases in which COST-FIRE is tractable.

COST-FIRE is a clear generalisation of FIRE, so the following is not difficult to justify. For completeness, we give a simple reduction.

**Theorem 3.** COST-FIRE *is NP-complete.*

*Proof*: Consider an arbitrary instance of FIRE and construct an instance of COST-FIRE with a cost function $c : V \to \mathbb{N}$ such that for all $v \in V$, $c(v) = 1$ and a budget $b = 1$. Clearly, this encodes FIRE as an instance that is a yes-instance of COST-FIRE if and only if the original instance of FIRE is a yes-instance, so COST-FIRE is NP-hard by reduction from FIRE. It is not difficult to justify that a certificate composed of a strategy $\sigma$ can be verified in time polynomial in the size of the instance. □

### 4.1 Sea Fan graphs

Sea Fan graphs are highly symmetric, sparse trees on which we show COST-FIRE remains NP-hard but FIRE can be solved in time polynomial in the size of the tree.

**Definition 4** (Sea Fan). A rooted tree $(F,r)$ is an $(f, \ell)$–Sea Fan graph if $F \setminus \{r\}$ is a forest of $f$ isomorphic trees called *fronds*, each composed of a path denoted $P$ of length $\ell$ and two vertices adjacent to the terminal vertex not adjacent to $r$ in $F$.

We reduce from the 2N2P-SAT problem, a variant of 3-SAT, to show that COST-FIRE is computationally hard on Sea Fan graphs by reduction. Berman et al. (referring to it as the $(3,2B)$-SAT problem) show this problem is NP-complete [16].

---

2N2P-SAT

**Instance:** A set of $n$ variables $\mathcal{V} = \{v_1, \ldots, v_n\}$ arranged into a logical formula $\mathcal{F}$ such that:

- $\mathcal{F}$ is in conjunctive normal form,
- each clause of $\mathcal{F}$ contains exactly 3 literals, and
- each variable in $\mathcal{V}$ appears in $\mathcal{F}$ exactly twice as a positive literal and twice as a negative literal.

**Question:** Is there a truth assignment for $\mathcal{V}$ that satisfies $\mathcal{F}$?

---

**Theorem 4.** COST-FIRE *is NP-complete on* $(f, \ell)$–*Sea Fan graphs rooted at a vertex r.*

*Proof*: We reduce from 2N2P-SAT: take as given an 2N2P-SAT instance $\phi$ of $n$ variables in $m$ clauses and let

$$\mathcal{V} = (v_1, v_2, \ldots, v_n),$$
$$\mathcal{C} = (C_1, C_2, \ldots, C_m)$$

be lexicographic orderings of the variables and clauses of $\phi$ respectively. Because each variable appears in the logical expression 4 times, and each clause contains 3 literals, $m = 4n/3$ (as observed by Berman et al. [16]). Create an instance of COST-FIRE on a $(2n, n+2m)$-Sea Fan denoted $(F,r)$ (as in Fig 2) with vertices labelled as follows.

1. For each variable $v_i \in \mathcal{V}$ for $1 \le i \le n$, select vertices to label with the positive and negative literals of $v_i$ respectively (called the 'literal vertices'):
   (a) the vertex in $V(F)$ on the path $P_i$ at distance $n+2m$ from $r$ is labelled $v_i^+$;
   (b) the vertex at distance $n+2m$ from $r$ on path $P_{n+i}$ is labelled $v_i^-$.
2. For each literal vertex, label its two neighbours of degree one $C_j$ and $C_k$ (the 'clause vertices'), the clauses in $\mathcal{C}$ in which the literal appears.

Let the target be $k = n+3m$, the budget be $b = 2$ and over the set of times $T = [1, n+m] \subset \mathbb{N}$ let the cost function $c : V \times T \to \mathbb{N}$ be

$$c(u,t) = \begin{cases} 2 & \text{if } u = v_t^+ \text{ or } v_t^-, \\ 1 & \text{if } u = C_{t-n}, \\ 3 & \text{otherwise.} \end{cases}$$

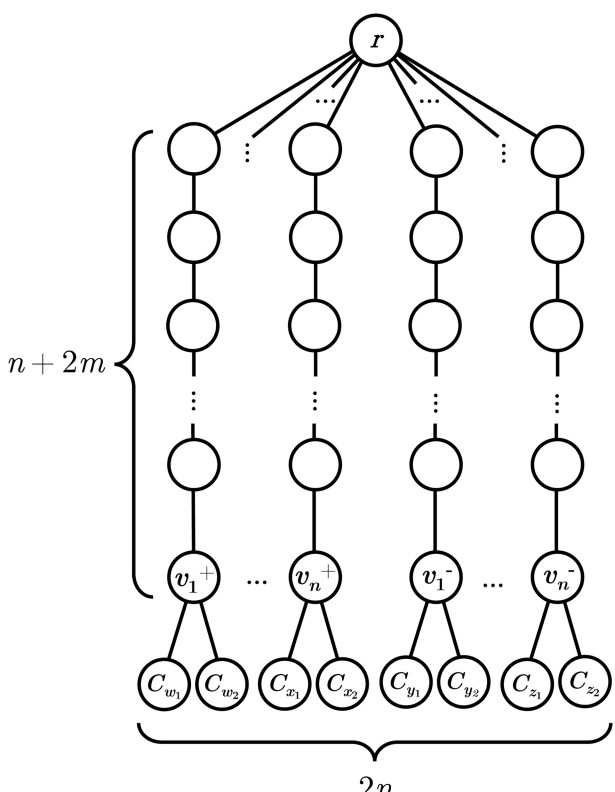

**Fig 2. Construction of a $(2n, n+2m)$-Sea Fan graph used in the reduction in the proof of Theorem 4; vertex labels correspond to variables and clauses in the 2N2P-SAT instance.**

Intuitively, this function ensures that one literal vertex for each variable can be defended at each time step up to $n$, and two clause vertices for each clause at each time step from $n + 1$ to $n + m$. For all three clause vertices labelled $C_j$ to be saved, at least one vertex labelled with a literal in $C_j$ must have been defended.

We now show that, if $\phi$ is a yes-instance, our instance of COST-FIRE is also a yes-instance. Given a satisfying truth assignment for $\phi$, let $\mathcal{L}$ be the set of satisfied literals. We deploy strategy $\sigma$ for the instance of COST-FIRE as follows. From time 1 to $n$, we defend the vertices corresponding to satisfied literals in $\mathcal{L}$. Since $\phi$ is satisfying, at least one of each set of three clause vertices is protected by time $n$, meaning $n + 2n$ vertices are protected so far. At each time $n < j \leq n + m$, defend the (up to) two vertices labelled $C_{j-n}$ not yet protected - there is only one to defend for clauses that contain the same literal twice, otherwise two. Saving these remaining clause vertices completes protection of all clause vertices for each clause in $\mathcal{C}$. Since there are $n$ undefended literals, and we have a satisfying truth assignment, $2n$ clause vertices have been directly defended in this phase. Recalling that $m = 4n/3$, in total $n + 2n + 2n = n + 4(3m/4) = n + 3m = k$ vertices have been saved.

To argue the opposite direction, we assume a 'yes' instance of COST-FIRE on the constructed Sea Fan graph with strategy $\sigma$. By cost function design, $\sigma$ saves $n$ literal vertices and $3m$ clause vertices, $2n$ of which by direct defence. Therefore, to save $n + 3m$ vertices, for every clause, at least one clause vertex is protected by defending a literal parent vertex. By cost function design, only one literal per variable can be defended. Hence, interpreting the literal vertices of $\sigma$ as truth values gives a satisfying assignment.

Having reduced from 2N2P-SAT, we have shown COST-FIRE is NP-hard. Verifying a certificate for any instance of COST-FIRE can be done in time polynomial in the size of the certificate by Theorem 3, hence COST-FIRE is NP-complete on $(f, \ell)$-Sea Fan graphs rooted at $r$. □

In contrast, we find that FIRE is polynomial-time resolvable on this same graph class.

**Theorem 5.** FIRE *is decidable in polynomial time on* $(f, \ell)$-*Sea Fan graphs when the fire starts at the root vertex.*

*Proof*: There exists an optimal strategy that always defends adjacent to burning vertices for instances of FIRE on trees [17]. On a Sea Fan graph, if we defend adjacent to fire on a frond, fire is contained in this frond. Because of the symmetry of the fronds, all defence choices adjacent to the fire on a frond will give an equivalent game state. Thus, there is an optimal defence that defends an open vertex adjacent to a burning vertex at each time, unless no such vertex exists in which case the process is over. Then, at each time $t$ an optimal strategy defends a vertex at distance $t$ until the end of the process. We depict an example of this strategy in Fig 3. This can be simulated in polynomial time, and thus FIRE can be decided efficiently on an $(f, \ell)$-Sea Fan graphs when the fire starts at the root vertex.

In our first defence, we save $\ell$ vertices on the path of the first frond and 2 vertices attached to the end of the path (so $\ell + 2$ overall). In the next turn, we defend $\ell + 1$ and so on. In the case of $f \leq \ell$, we defend $\ell - f + 3$ on the final turn before fire is contained (e.g. at time $\ell$, we defend a vertex at the end of a path, saving three vertices). Using this, when $f \leq \ell$, we save:

$$\sum_{\ell - f + 3}^{\ell + 2} i = f/2(2\ell + 5 - f).$$

If fire is not contained by turn $\ell + 1$ (that is, $f > \ell$), we defend one vertex of degree one. After this, any remaining fronds are completely burned. So, in the case of $f > \ell$, we save

$$(\ell + 2) + (\ell + 1) + 4 + 3 + 1 = \sum_{3}^{\ell + 2} i = \ell/2(\ell + 5) + 1.$$

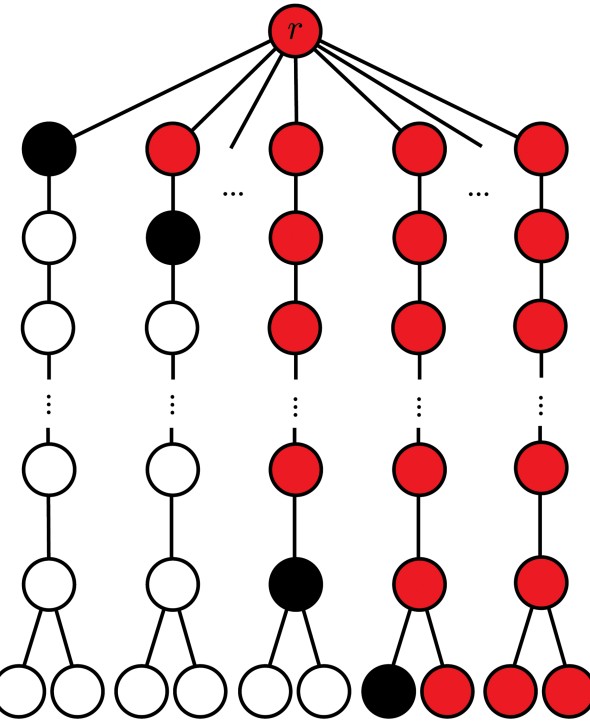

**Fig 3**. **Illustration of the strategy used in the proof of** Theorem 5 **(to show that FIRE on an (f, ℓ)-Sea Fan graph is tractable).** Open vertices are unfilled, burning are filled in red and defended are filled black.

Hence, an instance of FIRE on a $(f, \ell)$-Sea Fan is a yes-instance if and only if:

$$k \leq \begin{cases} f/2(2\ell + 5 - f) & \text{if } f \leq \ell, \\ \ell/2(\ell + 5) + 1 & \text{if } f > \ell. \end{cases}$$

$\square$

## 4.2 Complete graphs

We show that COST-FIRE is solvable on complete graphs, on which solving FIRE is tractable [4] but solving TEMP-FIRE remains NP-hard [13].

**Theorem 6.** *Instances of* COST-FIRE *on complete graphs on n vertices are solvable in $\mathcal{O}(n \ln n)$ time*

*Proof*: Consider an instance of COST-FIRE on a rooted complete graph $(K = (V, E), r)$. At time 0, fire breaks out at $r$ and every vertex becomes adjacent to a burning vertex, meaning any vertex not defended in the first turn will burn by the second. To maximise the number of vertices saved, we sort all vertices by cost at this time in $\mathcal{O}(n \ln n)$-time and defend in order of increasing cost (in $\mathcal{O}(n)$ time) until no other vertex can be defended without exceeding the budget. Hence, we can solve COST-FIRE on complete graphs in $\mathcal{O}(n \ln n)$ time overall. $\square$

Observe that this result holds for any graph in which the root is adjacent to all other vertices (e.g. star graphs rooted at the central node), of which complete graphs are an example.

## 4.3 Graphs of bounded path length

We say a graph is $P_\ell$-*free* if it contains induced paths of maximum length $\ell - 1$ (this class is usually called $P_k$-free, but $k$ is already a variable in FIRE); we take $\ell$ to be constant. FIRE on $P_\ell$-free graphs on $n$ vertices is solvable in $\mathcal{O}(n^\ell)$ time [5]. We show that COST-FIRE on this graph class is also tractable for fixed budget $b$ and maximum path length $\ell$.

**Theorem 7.** *Instances of* COST-FIRE *with budget* $b \in \mathbb{N}$ *on* $P_\ell$-free *graphs with n vertices and m edges can be solved in* $\mathcal{O}((n + m)n^{b(\ell-2)})$ *time.*

*Proof*: Observe that the length of the longest path bounds the lifetime of the instance, which is $\ell - 2$ for instances on $P_\ell$-free graphs. Thus, a valid strategy can defend up to $b(\ell - 2)$ vertices (when all vertices defended cost 1, we can defend at most $b$ vertices per turn). We generate potential defence sequences by enumerating sequences of distinct vertices of lengths up to $b(\ell - 2)$ in $\mathcal{O}(n^{b(\ell-2)})$ time. We simulate defence for each of these sequences as follows. For each turn, we obtain the cost of each vertex in the sequence (given the current problem state) and greedily defend until either all vertices in the sequence are defended or no more can be defended without exceeding the budget. We then validate the outcome by performing a depth-first search from $r$ in $\mathcal{O}(n + m)$ time to count all unburned (defended and open) vertices at containment of fire. We return a yes-answer if and only if there are at least $k$ such vertices. This algorithm runs in $\mathcal{O}((n + m)n^{b(\ell-2)})$ time. $\quad\square$

A problem with input parameter $k$ that can be solved by an algorithm that runs in time $n^{f(k)}$ for input size $n$ and a computable function $f$ is said to be in XP, meaning slicewise polynomial. We provide such an algorithm to prove Theorem 7 that solves COST-FIRE and is parameterised by the length of the longest path and the budget, leading to the following corollary. We prove a stronger result in Sect 4.5.

**Corollary 1.** COST-FIRE on graphs of maximum path length $\ell$ is in XP when parameterised by $\ell$ and the budget.

Notable examples of $P_\ell$-free graphs include *cographs* and *split graphs*; cographs are $P_4$-free and split graphs are $P_5$-free, so - similarly to FIRE [5] - this theorem implies the existence of algorithms for solving COST-FIRE with budget $b \in \mathbb{N}$ on cographs and split graphs on $n$ vertices, in particular, that run in $\mathcal{O}((n + m)n^{2b})$ and $\mathcal{O}((n + m)n^{3b})$ time respectively, although we conjecture that more efficient algorithms exist.

## 4.4 Trees

A natural question is whether there exists a generalisation of Theorem 2 for COST-FIRE. For instances of $b$-FIRE on a rooted tree in which each vertex has at most $b + 1$ children, there is an optimal strategy in which burning is restricted to a shortest path from the root to a vertex of degree $b + 1$ [9]. This is a generalisation of the well-known strategy to show that FIRE on trees where vertices have at most two children is tractable [4].

While COST-FIRE is not tractable on rooted trees in which each vertex has at most $b + 1$ children in general (which follows from Theorem 9 below), we define properties to re-couple the relationship between degree and budget that exists in FIRE, and made explicit in $b$-FIRE, to make COST-FIRE with these properties tractable on such trees. We refer to instances that possess this set of properties as PATHCONTAINABLE; the PATHCONTAINABLE property captures a restrictive but intuitive class of COST-FIRE instances in which we can achieve optimal containment by restricting the spread of fire to a single path in a tree, hence the term PATHCONTAINABLE. In particular, we find the shortest path to a leaf vertex (or a vertex with degree less than the maximum), inspired by the famous strategy previously mentioned [4]. This approximates real-world processes, such as in wildfire control where firefighters construct barriers just ahead of the fire line, or in epidemiological interventions that prioritise vaccinating individuals directly exposed to an infected contact.

**Definition 5** (PATHCONTAINABLE instance of COST-FIRE). Let $(T,r)$ be a rooted tree, $k$ and $b$ be positive integers and $c$ be a cost function. An instance of COST-FIRE is PATHCONTAINABLE if:

1. For any open (unburned and undefended) vertex $u$ of $T$ adjacent to a burning vertex and any open vertex $w$ not adjacent to a burning vertex, $c(u) \leq c(w)$.
2. For any burning vertex $v$ with $x$ open neighbours, we have sufficient budget to defend any subset of $x–1$ of them.

Importantly, this definition is independent of strategy played, and so is very restrictive. Nonetheless, there are several intuitive, state-dependent cost functions that are natural in common applications of FIRE and variants (e.g. disease modelling) that are PATHCONTAINABLE. For instance, the functions in examples 5 and 6, which are related to proximity to fire, with an appropriate choice of budget.

We now prove that solving PATHCONTAINABLE instances of COST-FIRE is tractable, even on state-dependent and temporal cost functions, using an approach inspired by Fomin et al. [5].

**Lemma 1.** *For* PATHCONTAINABLE *inputs of* COST-FIRE*, there is an optimal solution in which each defended vertex is adjacent to a burning vertex when defended.*

*Proof*: This is trivial when we can defend all children of the root as the first defence, as the process would then end. Hence, assume that the budget and cost function are such that we can defend precisely all-but-one children of the root. Suppose for a contradiction that there is no optimal solution in which all defended vertices were adjacent to a burning vertex when defended and let $(T,r)$ be a graph with fewest vertices for which this is the case and thus a minimal counterexample.

Suppose there is an optimal strategy $\sigma$ in which the vertices defended on the first turn are all neighbours of $r$, labelled $x_1, \dots, x_{\Delta-2}$. Then, the rooted subgraph $T[x_{\Delta-1}]$ is a smaller counterexample, a contradiction. Hence, we assume the set of vertices defended by $\sigma$ in the first turn includes at least one vertex $v$ not adjacent to the burning root. At the time of the first defence, the ancestor $u$ of $v$ adjacent to the burning root is either defended or open. If $u$ is defended in the first turn, it is unnecessary to then defend $v$, so $\sigma$ is not optimal; if $u$ is open, defending it rather than $v$ would save at least one more vertex and neither $u$ nor $v$ catch fire. Thus, in both cases, a strategy that defends beside fire performs at least as well, a contradiction. □

Given an instance of COST-FIRE on a tree, we call a vertex *defensible* with respect to a strategy $\sigma$ in a certain turn if we can defend all of its open neighbours in that turn.

**Lemma 2.** *For* PATHCONTAINABLE *inputs of* COST-FIRE*, there exists an optimal solution in which only a path of minimum length to a defensible vertex burns.*

*Proof*: By Lemma 1, there is an optimal strategy in which each vertex defended was adjacent to a burning vertex when defended. Given the input restrictions, we can defend any set of all-but-one children of a burning vertex, so clearly at most one vertex burns at each level of the tree. Since the parent of every burning vertex is burning, the burning vertices induce a path in the tree. □

**Theorem 8.** *We can decide* PATHCONTAINABLE *inputs of* COST-FIRE *on trees with $n$ vertices and $m$ edges in $\mathcal{O}(n+m)$ time.*

*Proof*: For PATHCONTAINABLE inputs of COST-FIRE, by Lemma 2 there exists an optimal solution that only burns a path. We can use an adapted breadth-first search to find the shortest path to a closest defensible vertex $v$ in the graph in time $\mathcal{O}(n+m)$ and then simulate the instance with the strategy of burning this path in at most $n$ turns. □

An obvious question is whether both PATHCONTAINABLE properties are required for tractability. The following gives us that property 1 alone is not sufficient, whether 2 alone is sufficient we leave open.

**Theorem 9.** *Solving instances of* COST-FIRE *on rooted trees $(T = (V, E), r)$ that satisfy* PATHCONTAINABLE *property 1 but not 2 is NP-complete.*

*Proof*: Consider an arbitrary instance of $b$-FIRE on a budget $b$, a rooted tree $(T = (V, E), r)$ of maximum degree $b + 2$ such that $\deg(r) = b + 2$ and an integer $k$ - solving this is know to be NP-complete [18]. We construct an instance of COST-FIRE with budget $b$, rooted tree $(T,r)$ and integer $k$ and a cost function that assigns a cost of one to all vertices. This clearly encodes instances of $b$-FIRE as an instance of COST-FIRE, showing that solving such instances of COST-FIRE is NP-hard. To show this encoding satisfies PATHCONTAINABLE property 1 and not 2, first observe that PATHCONTAINABLE property 1 is clearly satisfied as all costs are one. Note that the root is of degree $b + 2$ and we can defend up to $b$ vertices per turn, so we can defend all-but-two vertices adjacent to the root when it burns, meaning PATHCONTAINABLE property 2 is not satisfied. Verification of the certificate in time polynomial in the size of the instance is straightforward. Hence, COST-FIRE on such restricted inputs is also NP-complete. □

## 4.5 Parameterised tractability

A problem is fixed parameter tractable (FPT) with respect to a parameter $k$ if it can be solved by an algorithm with running time $f(k)n^{O(1)}$, where $f$ is a computable function and $n$ is the size of the input. Compare this to the class of slice-wise polynomial problems (XP) referenced in Sect 4.3, which are problems that admit algorithms that run in time $n^{f(k)}$. Corollary 1 states that COST-FIRE on static cost functions is in XP parameterised by the length of the longest path and budget. Using a result due to Arnborg et al. [19] and inspired by Bazgan et al. [20], we express COST-FIRE in a variant of Monadic Second-Order logic (MSO) to prove it is FPT by the sum of treewidth, budget and problem lifetime.

MSO is the fragment of second-order logic produced by limiting second-order quantification to monadic (single argument) predicates. This system permits: variables for (sets of) vertices and (sets of) edges; the usual logical connectives $\neg, \vee, \wedge, \implies$ and $\iff$; quantifiers $\forall$ and $\exists$; and well-formulated predicates. We use an extension of MSO called Extended MSO (EMSO), which allows us to count the size of sets that are quantified by second-order variables.

We rely on the following generalisation from MSO to EMSO of Courcelle's Theorem [6] due to Arnborg et al. [19]. A more general version of the following was presented in a survey by Langer et al. [21].

**Theorem 10.** *[19] Let $P$ be an EMSO-definable problem. We can solve $P$ on graphs $G = (V, E)$ of order $n = |V|$ with treewidth at most $w \in \mathbb{N}$ in time $\mathcal{O}(f_P(w) \cdot n)$.*

To use this theorem, we produce a EMSO expression for COST-FIRE with size that is a function of budget and maximum time step, thereby showing COST-FIRE is in FPT parameterised by these and the treewidth of the input graph.

**Theorem 11.** COST-FIRE *is in FPT when parameterised by budget, maximum time step and treewidth of the input graph.*

*Proof*: Consider an arbitrary instance of COST-FIRE on a rooted graph $(G = (V, E), r)$, cost function $c$ and positive integer budget $b$, with maximum time step $T \in \mathbb{N}$. Let $\sigma$ be a strategy, a sequence of vertices ordered by defence, up to time $T$. Denote by $\sigma_i$ the vertices defended up to time $i$. Denote by $F_i$ the set of vertices burning at time $i$ under this strategy.

We define the following generalisations of predicates defined by Bazgan et al. [20]. NextToBurn returns true when, given $\sigma_i$ and $F_{i-1}$, the defended vertices up to $i$ and burning vertices up to time $i$, $F_i$ contains the previously burning vertices and those that catch fire at time $i$. For this, we define a predicate $A : V \times V \to \{\texttt{true}, \texttt{false}\}$ such that $\forall u, v \in V$, $A(u, v) = \texttt{true}$ if and only if $u$ and $v$ are adjacent in $G$.

$$\text{NextToBurn}(F_{i-1}, F_i, \varsigma_i) \coloneqq$$
$$\forall v \in V, v \in F_i \iff v \in F_{i-1} \vee \exists u \in V(u \in F_{i-1} \wedge A(u, v) \wedge v \notin \varsigma_i)$$

The next predicate evaluates to true when all vertices in a set $S$ are 'saved' at the end of the process under a strategy $\sigma$, i.e. no vertex in $S$ is burning and each neighbour of each vertex in $S$ is either saved or defended.

$$\text{Saved}(S, F_T, \sigma) \coloneqq \forall u \in V(u \in S \implies (u \notin F_T \wedge \forall v \in V(A(u, v) \implies v \in S \cup \sigma)))$$

Since the process ends when fire can no longer spread, this is sufficient to verify that fire cannot spread to the vertices in this set.

The final predicate ensures that $\sigma$ is a valid defence strategy.

$$
\text{ValidStrat}(b, \sigma) := \left( \bigwedge_{1 \leq i \leq T} \sum_{j=1}^{r_i} c(d_i^j, i, S_{V,i}) \leq b \right)
$$

$$
\wedge \bigwedge_{\substack{1 \leq i \leq T \\ 1 \leq j \leq r_i}} \left( \left( \bigwedge_{\substack{1 \leq k \leq i \\ 1 \leq \ell \leq r_k}} (i = k \wedge j = \ell) \vee (d_i^j \neq d_k^\ell) \right) \wedge (d_i^j \notin F_i) \right)
$$

$$
\wedge \, \forall \sigma_i \in \sigma \left( \bigwedge_{v \in \sigma_i} v \notin F_T \wedge \left( \sum_{v \in \sigma_i} c(v, i, S_{V,i}) \leq b \right) \wedge \forall \tau \in \sigma \left( \bigwedge_{\substack{d \in \sigma_i \\ d' \in \tau}} d \neq d' \right) \right)
$$

Using these predicates, we can express COST-FIRE in EMSO.

$$
\text{CostFire}(k, b) := \exists \sigma \, \exists F_0, \dots, F_T, \exists S, \, \forall u \in V, (u \in F_0 \iff u = r)
$$

$$
\wedge \bigwedge_{1 \leq i \leq T} \text{NextToBurn}(F_{i-1}, F_i, \sigma)
$$

$$
\wedge \, \text{ValidStrat}(b, \sigma)
$$

$$
\wedge \, \text{Saved}(S, F_T, \sigma) \wedge (|S| \geq k)
$$

Since we have expressed an arbitrary instance of COST-FIRE in EMSO, and this expression is clearly of constant length, by Theorem 10, COST-FIRE is in FPT when parameterised by the treewidth of $G$, the budget $b$ and by the maximum time step. □

## 5 Experimental methods

Motivated by the formal hardness of COST-FIRE and an interest in the behaviour of the game in practice, we investigate the problem empirically. We report on the number of vertices saved by different heuristics on various cost functions and budgets on random and real-world derived graphs. Graphs were chosen for a range of metrics and thus potential modelling applications.

We wrote a Python script to simulate, analyse and store results of instances of COST-FIRE on given graph types, which is available at https://github.com/Ethan-CS/CostFnFire. The script can be used with random graphs generated with the Networkx library [22] or with graphs from data files.

Heuristics iteratively select vertices for defence each turn, until either the sum of the costs of the selected vertices meets the budget limit or no remaining open vertex can be defended within the remaining budget. We implemented four heuristics for comparison; each was evaluated alone and with the two other non-random strategies to break ties in turn in cases where more than one vertex could be selected by the primary heuristic. The heuristic defence strategies implemented are:

1. **Random** - select vertices randomly,
2. **Degree** - defend in order of highest to lowest vertex degree,
3. **Threat** - defend in order of least to greatest distance from fire, and
4. **Cost** - defend in order of least to greatest cost.

Importantly, these heuristics have no guarantees of optimality, but are common in simulation approaches to firefighter problems - see e.g. work by García-Martínez et al. [23].

The cost functions recalculate costs for each vertex at the start of every turn. We implemented the following functions:

1. **Uniform:** all vertices cost 1 at all times.
2. **Binary hesitation:** vertices cost 2 with probability 0.297, 1 otherwise (to mirror the vaccine hesitancy rate found by Pourrazavi et al. [3]).
3. **Uniformly random:** costs uniformly randomly sampled from the range [1,5]
4. **Threat with high stochasticity:** distance from fire plus an integer sampled uniformly randomly from [–3,3].
5. **Threat with low stochasticity:** distance from fire plus an integer sampled uniformly randomly from [–1,1]

For very dense graphs, threat is very likely to be high for a given vertex (i.e. distance from fire is generally low). Since we define cost functions so that costs are always strictly greater than 0, in the high stochastic case (an added integer uniformly sampled from [–3,3]) we expect costs to be low. Thus, on dense graphs, we expect heuristics to perform similarly with uniform and highly-stochastic threat-based cost functions.

For each graph, and for various budgets, we plot the average (median) of 50 trials with an envelope (shaded area) representing the 95% confidence interval from this average. For each random graph class, we select one results plot to present in each section, with others provided in supplementary information as they are qualitatively similar. We detail the graphs used for experiments in Sect 5.2 and results obtained in Sect 6.

## 5.1 Experimental design

We generate 50 trials, each a combination of: a graph type, a cost function and an heuristic strategy. In the case of random graphs, we generate a new graph on 100 vertices and a range of generation parameters for each trial; we use the same graph to assess each cost function-heuristic pair. The outbreak vertex was selected randomly for each trial. We stored results, including protection strategies, cost mappings and overall performance metrics, for later verification, analysis and plotting. A time-out of 120 minutes was set, which was reached usually in cases of uniformly-random and threat-based with high stochasticity by cost-based heuristics as this amounts to random selection. We expect random selection to extend instance lifetime on average, since effort is not targeted towards containment.

## 5.2 Graphs used

We studied Cost-Fire simulations on two real-world graphs to see how the problem performs as a model for disease on real contact graphs. We chose two contact graphs, one of lizards and another of raccoons, which we selected for their contrasting graph structures and public availability.

We also studied heuristic performance on four random graph classes: Erdős-Rényi, Barabási-Albert, geometric and *n*-regular random graphs. These were selected to study how heuristics perform across more controlled changes in graph structure (e.g. by increasing the probability used for Erdős-Rényi graph generation, we expect graph density to increase). Across these random graph classes, cost-based heuristics (particularly with ties broken on highest threat) outperformed the others in most cases. This performance was followed closely - and occasionally slightly outperformed by - the 'reverse' of this heuristic, threat-based with cost-based tie breaks.

**5.2.1 Sleepy lizards.** Bull et al. [24] studied a group of Australian scincid 'sleepy' lizards (*Tiliqua rogosa*) to investigate the spread of *Salmonella enterica* bacteria within the group.

They concluded that infected lizards were far more likely to have contracted the bacteria through contact with another lizard than from some common environmental source [24], which motivated our application of Cost-Fire as a model of disease on this population. We were further motivated to study Cost-Fire on this graph due to the unusual social

behaviours of this species: sleepy lizards exhibit non-random interaction behaviours, including monogamous pairings after extended courtships of six-to-eight weeks [25].

By studying a population of 60 lizards (30 male, 30 female) living in an area of 1.5 km$^2$ in Southern Australia, Bull et al. [24] produced a social interaction graph available at the Network Data Repository [26], which we visualise and plot degree rank of in Fig 4. To produce this graph, researchers attached data logger units to the lizards in September 2010, which recorded GPS locations every 10 minutes for 120 days. An edge exists between two lizards to represent social contact if the two incident lizards were within 2 m of each other for at least one of the 10 min GPS readings. We do not use the edge weights, which indicate the ratio of frequency of associations [24]. The resulting graph contains 60 vertices connected by 318 edges, its density is 0.180 and the maximum, mean and minimum degrees of vertices in the graph are 23, 10.6 and 2 respectively. Assortativity is positive, at 0.350, and clustering is fairly high, at 0.567.

The degree rank plot in Fig 4 indicates this interaction graph has some scale-free characteristics, as the trend is slightly curved, but is unlikely to be strictly scale-free. There are few vertices in the graph with low degree and many have fairly similar degree. Due to the high assortativity of this graph, we expect degree-based heuristics to perform poorly in simulations.

**5.2.2 Raccoons.** To model seasonal rabies outbreaks and vaccine efficacy over different levels of coverage, Reynolds et al. used proximity detection collars to construct a contact graph of suburban raccoons (*Procyon lotor*) from an area of 0.2 km$^2$ in Illinois [27].

The resulting contact graph on 24 vertices and 226 edges has a very high density of 0.819. The average degree of a vertex is 18.8, with a maximum degree of 23 and a minimum of 7. This, combined with a clustering coefficient of 0.903, indicates a highly interconnected graph. We see a fairly gentle slope in the degree rank plot in Fig 5, indicating a more

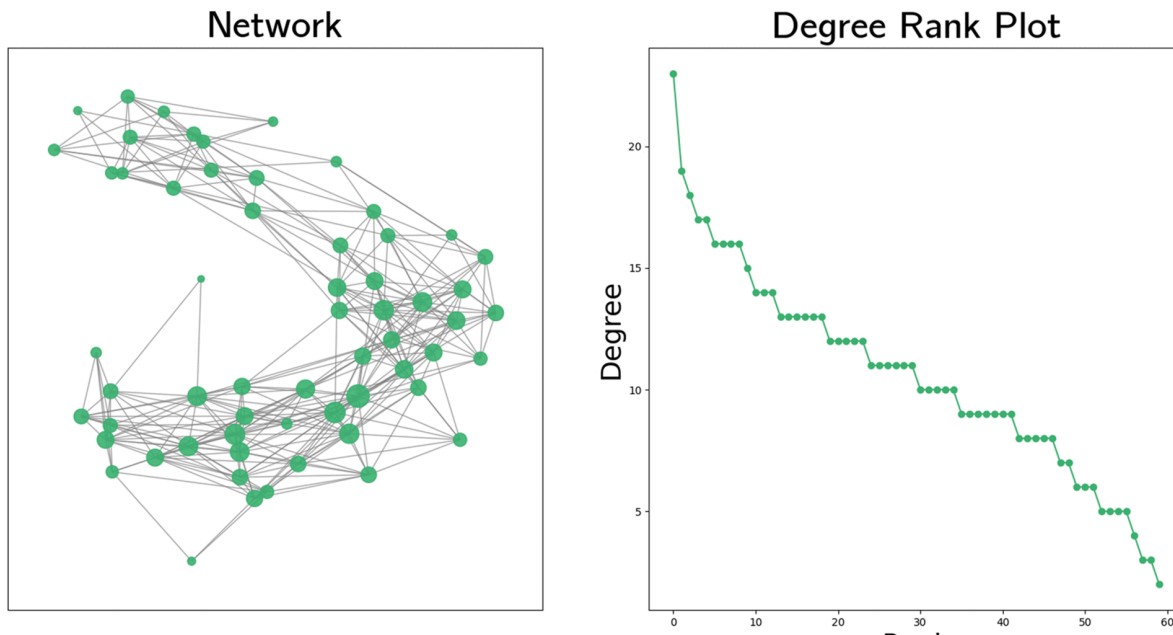

**Fig 4**. **Visualisation and degree rank plot of a social interaction graph of sleepy lizards due to Bull et al. [24].** In the graph visualisation, larger vertices indicate higher degrees.

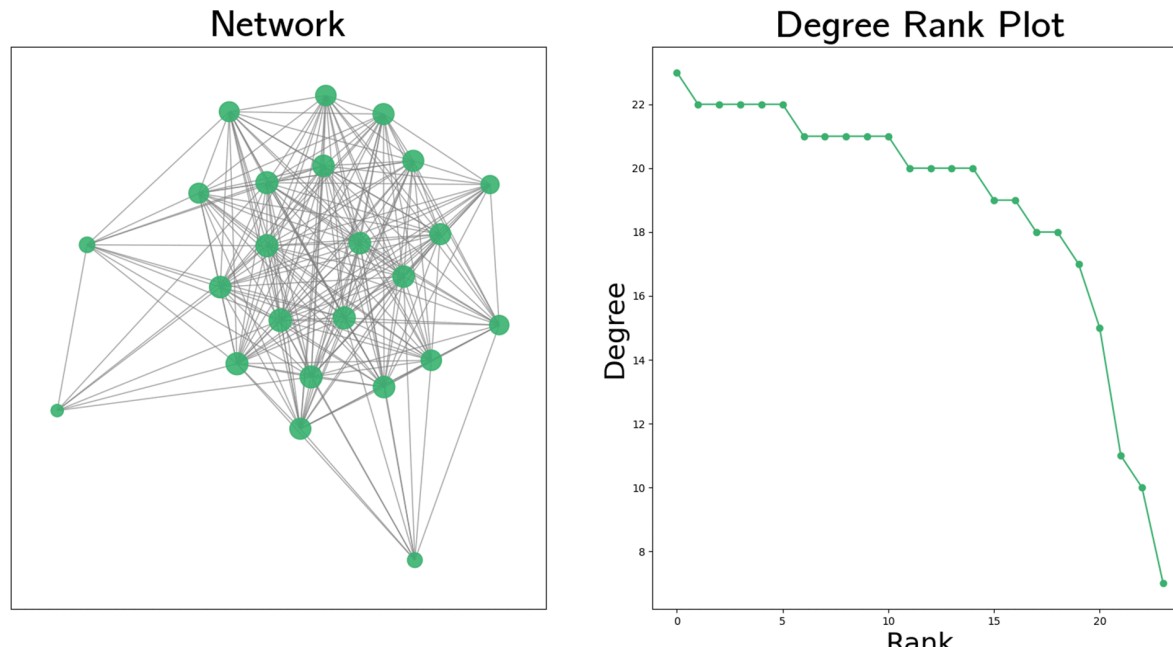

**Fig 5**. **Visualisation and degree rank plot of the raccoon contact graph constructed by Reynolds et al. [27] used to simulate heuristics.** Larger vertices indicate higher degrees.

even distribution of edges than the sleepy lizard contact graph. The graph has negative assortativity (-0.133), which suggests high-degree raccoons tend to connect with low-degree raccoons slightly more often than with other high-degree raccoons, which could slow disease propagation. For simulations of Cost-Fire on this graph, we expect degree-based heuristics to be less effective than in more sparse or scale-free graphs, such as the sleepy lizard graph. We chose this graph because, in the original research, Reynolds et al. used the contact graph to model outcomes with varying vaccination coverage. In our experiments, this is studied as varying budget.

**5.2.3 Erdős-Rényi.** Erdős-Rényi random graphs are generated on $n$ vertices by connecting each pair of vertices with given probability $p$ [28]. We chose this graph class as a 'baseline' for comparison that allows us to study the impact of varying graph density on the heuristic performances.

**5.2.4 Barabási-Albert.** Barabási and Albert developed an algorithm to randomly generate scale-free graphs that is given as input a number of vertices $n$ and a number of edges $m$ to attach [29]. The Barabási-Albert model generates a random graph by iteratively introducing $n$ vertices and randomly attaching each vertex with $m$ edges to existing vertices with preference for high-degree vertices. We selected this model for experiments as it produces scale-free graphs with power-law degree distributions, which is seen in many observed interaction graphs [30].

**5.2.5 Clustered power law.** Clustered power-law graphs combine scale-free (power-law) degree distributions with clustering coefficients. These graphs are randomly generated using an algorithm due to Holme and Kim [31], which extends the Barabási and Albert algorithm [29] with a *triangulation* step. That is, after each new edge is attached, another edge is created with probability $p$ between two of the neighbours of the newly connected vertex, forming a triangle [31]. This can result in higher average clustering, making it suitable for generating graphs with more heterogeneity and local clustering than the Barabási-Albert model [31].

**5.2.6 Watts-Strogatz.** The Watts-Strogatz model, due to Watts and Strogatz [32], generates small-world graphs, which are graphs with high clustering coefficients and short average path length. We begin with a ring lattice on $n$ vertices in which each vertex is connected to its $k$ closest neighbours and rewire each edge with given probability $p$ to a random other vertex. Setting the rewiring probability $p$ to 0 results in a regular ring lattice with high clustering and long path lengths, setting it to 1 results in a random graph with low clustering and short path lengths. For intermediate values of $p$, we reliably observe the small-world phenomenon [32], which is why we chose this model.

**5.2.7 Caveman.** Caveman graphs on $\ell$ cliques (caves) of size $s$ are connected graphs, constructed by rewiring a single edge in each clique to a vertex in another clique, forming a central cycle between cliques. We chose caveman graphs as they exhibit small-world properties by design but maintain global connectivity [33].

**5.2.8 Geometric.** A random geometric graph is generated by uniformly randomly placing vertices in the unit cube and connecting vertices if the distance between them is less than a specified radius [34]. We considered radii from 0.1 to 0.5; for radii less than 0.2, most of the vertices are saved regardless of heuristic due to the low expected density. We selected this class of random graph because of the high likelihood of cluster separation, unlike Erdős-Rényi or Barabási-Albert random graphs.

**5.2.9 Regular.** An $r$-regular graph is a graph in which each vertex has degree $r$; random $r$-regular graphs are selected uniformly randomly from the set of all possible $r$-regular graphs, although practical generation of $r$-regular graphs is non-trivial [35]. We use NetworkX for random graph generation, which generates random $r$-regular graphs using an algorithm due to Steger and Wormald [36]. Random regular graphs were selected to study COST-FIRE on graphs with regular degree; we expect degree-based heuristics to perform similarly to random defence for these graphs, since degree is uniform.

# 6 Experimental results and discussion

The median numbers of vertices saved at containment of fire under various heuristics on the sleepy lizard contact graph are plotted in Fig 6.

This figure shows that cost-based heuristics performed well across all implemented cost functions. As expected, we observe some similar performance from other heuristics when costs approach homogeneity (uniform and threat with high stochasticity, as reasoned in Sect 5). Under uniformly sampled and binary hesitation cost functions, threat and degree

**Fig 6. Plots of median numbers of vertices saved by various heuristics on an interaction graph of 60 sleepy lizards.**

heuristics perform similarly to random. This is also expected, since the graph is very dense (with density 0.180 and mean degree 10.6).

For threat-based costs with low stochastic variance, cost-based heuristics perform well, particularly when used as a tie break. In some cases, this heuristic outperforms the primarily cost-based heuristic in some cases. Degree-based heuristics consistently perform similarly to or worse than random defence.

We plot the results of experiments on the raccoon contact graph in Fig 7. As in lizard graph simulations, when costs are uniformly random and binary according to hesitancy, heuristics based or breaking ties on costs consistently outperform the other heuristics. However, on the raccoon contact graph, the difference in median numbers of vertices saved narrows between the random and cost-based heuristics at higher budgets. This is expected, because this number approaches the total number of vertices in the graph that can be saved (i.e. one less than the size of the graph).

When costs are threat-based with high stochastic variance, the picture of heuristic performance on the raccoon contact graph is very similar to the case of uniform costs, i.e. all perform similarly well, including random selection. As was the case for simulations on the lizard contact graph, this is because of the structure of the interaction graph - the graph is small and has a diameter 2 and only 50 of 276 pairs of vertices have minimum distance 2 in the graph - and the way variance is handled, discussed in Sect 5. In the lower variance case, lower homogeneity of costs mean that random selection tends more towards sub-optimal performance, so cost- and threat-based heuristics start to outperform random selection, particularly with higher budgets.

Moving onto random graphs, we plot simulation results on Erdős-Rényi graphs generated on 100 vertices with probability 0.05 in Fig 8 and with probability 0.25 in Fig 9. We provide simulation results for these graphs on the intermediate probabilities 0.10 and 0.15 in S1 Fig and S2 Fig respectively.

As in the real-world contact graph simulations, we see degree-based heuristics performing worse than the other heuristics on Erdős-Rényi graphs, sometimes worse than random. This is as expected, due to the expected uniformity of degrees in Erdős-Rényi graphs. Cost-based defences perform most consistently well across the cost functions and budgets.

Results of simulations on Barabási-Albert graphs on 100 vertices are presented for $m = 2$ in Fig 10 and $m = 10$ in Fig 11. Results for $m = 1, 3$ and 5 are given in S3 Fig, S4 Fig and S5 Fig respectively.

Degree-based heuristics again performed worst, although not as poorly (i.e. more similarly to random selection) as on Erdős-Rényi graphs. This is expected, because of the scale-free nature of Barabási-Albert on which degree-based

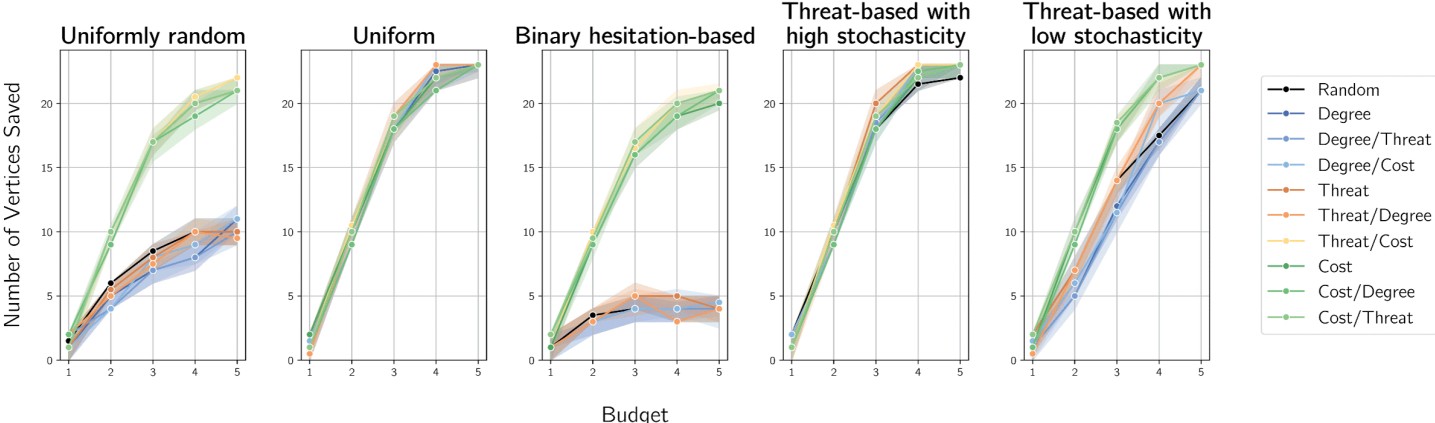

**Fig 7**. Plots of median numbers of vertices saved on an interaction graph of 24 raccoons by various heuristics.

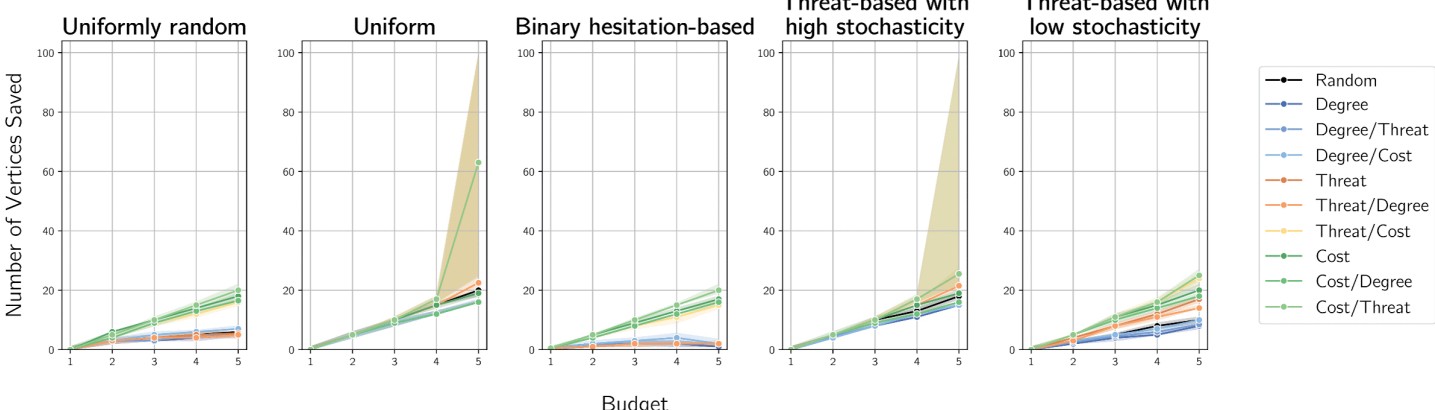

**Fig 8**. Plots of median numbers of vertices saved on randomly generated Erdős-Rényi graphs on 100 vertices with generation probability 0.05.

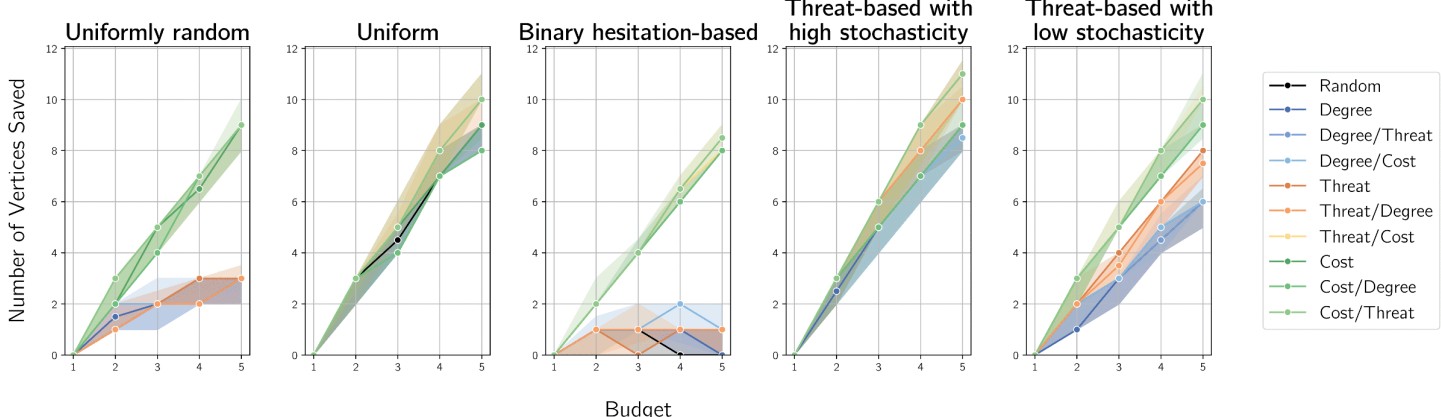

**Fig 9**. Plots of median numbers of vertices saved on randomly generated Erdős-Rényi graphs on 100 vertices with generation probability 0.20.

defences are more likely to prove effective. Compare this to the difference in degree-based heuristic performance between the lizard and raccoon graphs - in the former, which is more scale-free, degree-based heuristics performed worst but less poorly than in the latter, less scale-free case. Reassuringly, when cost correlates to threat, threat- and cost-based heuristics perform similarly. Cost-based strategies perform well with most cost functions, which becomes clearer as $m$ increases. We also notice differences in heuristic performance with uniformly random costs, particularly in the higher numbers of vertices saved by cost-based heuristics, in contrast to preliminary results in which all heuristics performed similarly poorly.

In Figs 12 and 13, we see very similar results on clustered power-law graphs on 100 vertices (with 2 and 5 edges added per vertex respectively) to the results on Barabási-Albert graphs. This is expected, as clustered power-law graphs are scale-free and the Barabási-Albert model produces scale-free-like random graphs. We give plots for simulation results on clustered power-law graphs on 100 vertices with 3 and 4 edges added per vertex in S6 Fig and S7 Fig respectively.

## Heuristic performance comparison for Barabási–Albert graphs with input edges 2

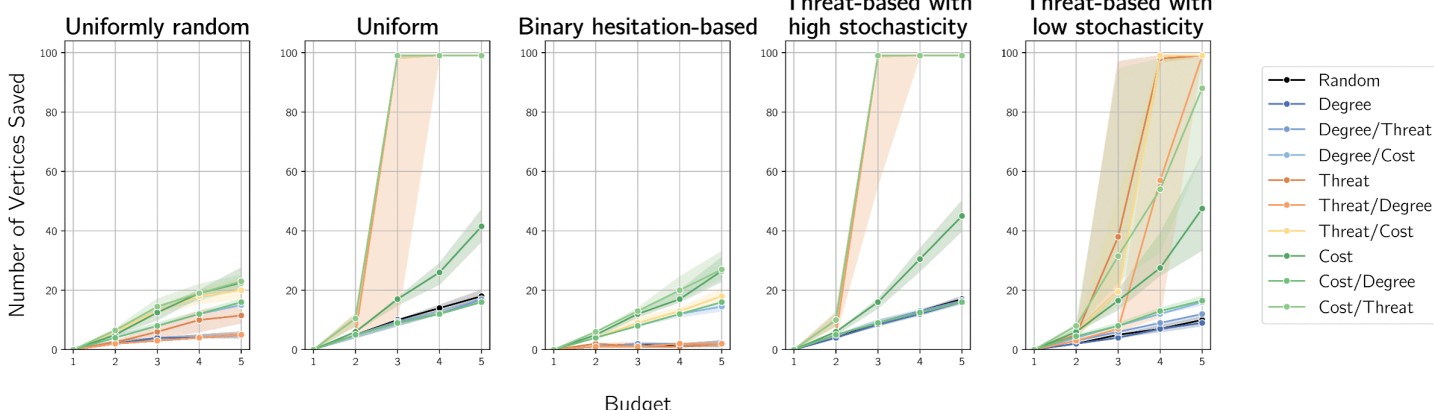

**Fig 10**. **Plots of median numbers of vertices saved on randomly generated Barabási-Albert graphs on 100 vertices with 2 edges added per vertex.**

## Heuristic performance comparison for Barabási–Albert graphs with input edges 10

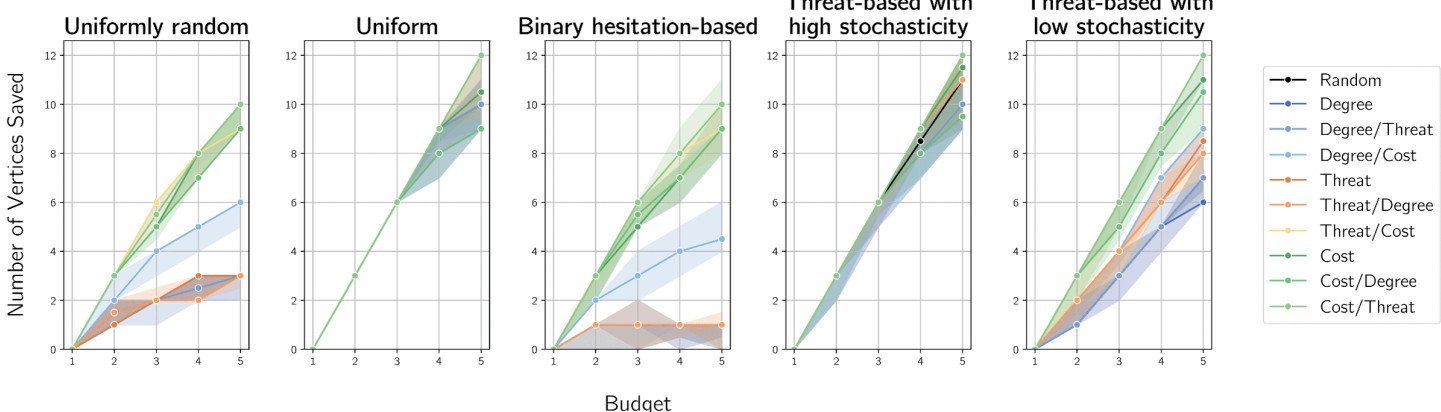

**Fig 11**. **Plots of median numbers of vertices saved on randomly generated Barabási-Albert graphs on 100 vertices with 10 edges added per vertex.**

We plot results on graphs generated by the Watts-Strogatz model on 100 vertices with mean degrees 4 and 10 in Figs 14 and 15 respectively. These plots show another improvement in threat-based heuristics compared to previous results. This may be expected, due to the small-world properties of such graphs. Nonetheless, cost-based heuristics often outperform all others, particularly with ties broken on degree. Results for Watts-Strogatz graphs on 100 vertices with mean degrees 6 and 8 are given in S8 Fig and S9 Fig respectively.

We also considered caveman graphs, another generation model that produces small-world graphs. Here, threat seems to have the greatest relative performance we have yet observed, particularly as a tie-break for a cost-based heuristic (and as the primary heuristic with ties broken on costs). Since this graph is so well connected, fire must be contained quickly as possible lifetime is naturally limited. Hence, defending as many as possible (by selecting low-cost vertices) as close

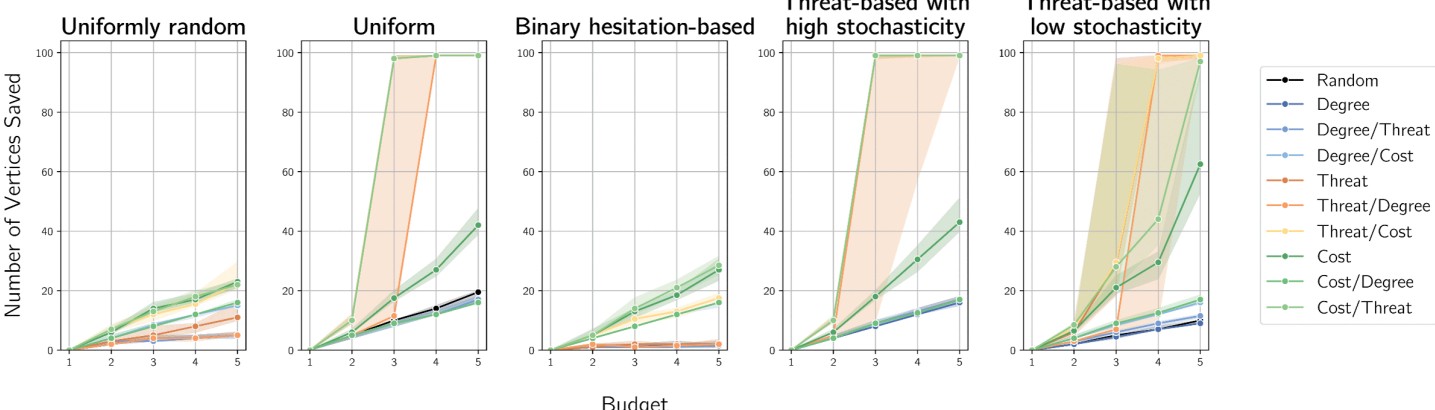

**Fig 12**. **Plots of median numbers of vertices saved on randomly generated clustered power-law graphs on 100 vertices with 2 edges added per vertex.**

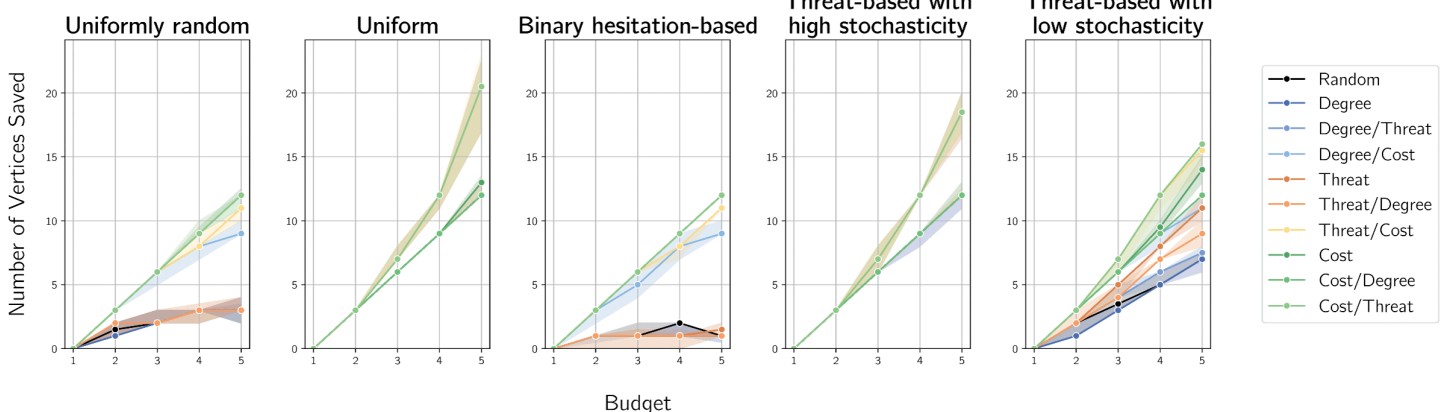

**Fig 13**. **Plots of median numbers of vertices saved on randomly generated clustered power-law graphs on 100 vertices with 5 edges added per vertex.**

as possible to fire (by breaking ties on proximity to fire) is a sound strategy, a conjecture supported by the plots in Figs 16 and 17 of caveman graphs on 100 vertices arranged in 10 and 20 cliques respectively.

Plots of median vertices saved by the heuristics for geometric graphs on 100 vertices with radii 0.25 and 0.5 are provided in Figs 18 and 19 respectively.

On geometric graphs, threat-based heuristics outperform cost-based heuristics when costs are uniform and they perform similarly with costs close to uniformly random (high stochasticity). In all other cost assignment cases, cost-based heuristics outperform all others.

Finally, we plot the median numbers of vertices saved on randomly generated 3-regular and 8-regular graphs on 100 vertices in Figs 20 and 21. We provide plots for 4- and 6-regular graphs in S11 Fig and S12 Fig respectively.

### Heuristic performance comparison for Watts-Strogatz graphs with mean degree 4

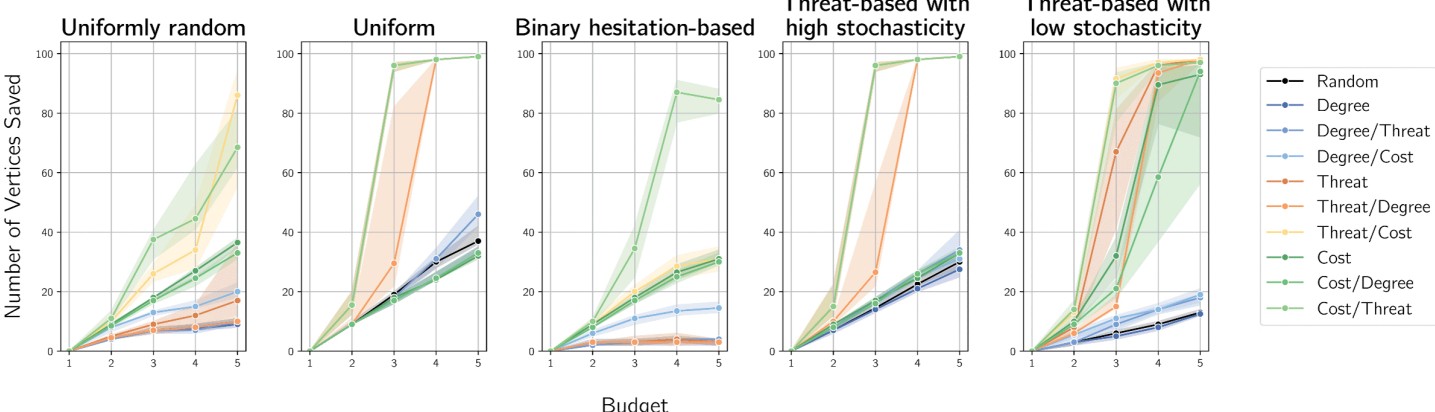

**Fig 14. Plots of median numbers of vertices saved on Watts-Strogatz graphs on 100 vertices with mean degree 4.**

### Heuristic performance comparison for Watts-Strogatz graphs with mean degree 10

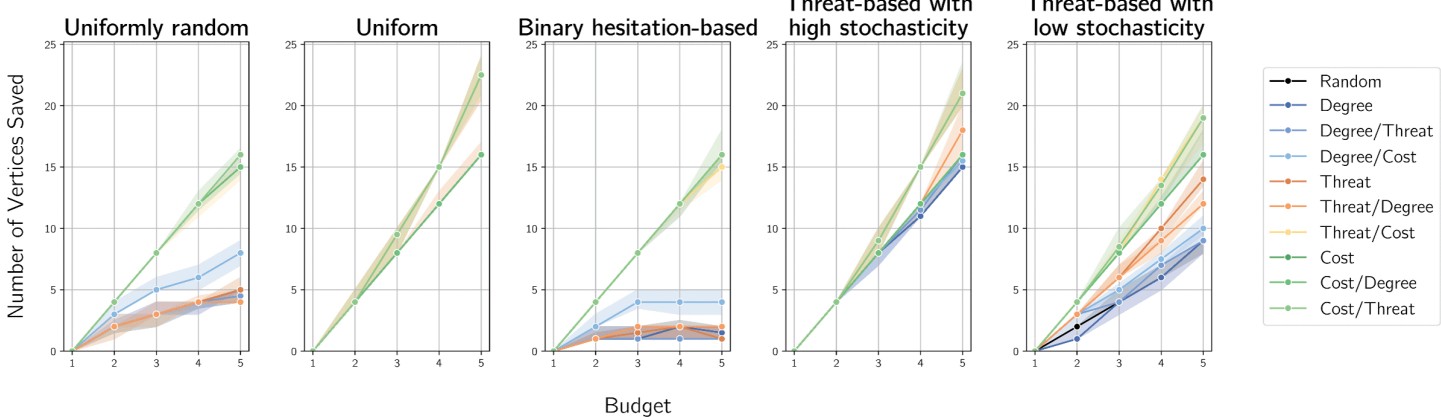

**Fig 15. Plots of median numbers of vertices saved on Watts-Strogatz graphs on 100 vertices with mean degree 10.**

On *r*-regular graphs, defending based on costs with ties broken on threat performs very well, particularly as budget increases. We observe a sharp jump in the uniform and near-uniform (high stochasticity) cases when the budget is equal to *r* in all cases. The only heuristic that performs similarly the cost with threat-based tie breaking heuristic is its 'reverse,' threat-based defence with ties broken on costs.

## 7 Conclusion

We have introduced COST-FIRE, a new variant of FIRE that includes dynamic cost functions for vertices, allowing us to define costs that vary in time or in response to burning in the graph. Like FIRE, we prove that COST-FIRE is NP-complete in general (Theorem 3), even on Sea Fan graphs on which FIRE is tractable. We showed that we can solve COST-FIRE in time polynomial in graph size when the input graph is complete (Theorem 6) and, with further input restrictions, a tree (Theorem 8). We also gave an algorithm that solves instances of COST-FIRE with budget *b* on graphs on *n* vertices and *m* edges

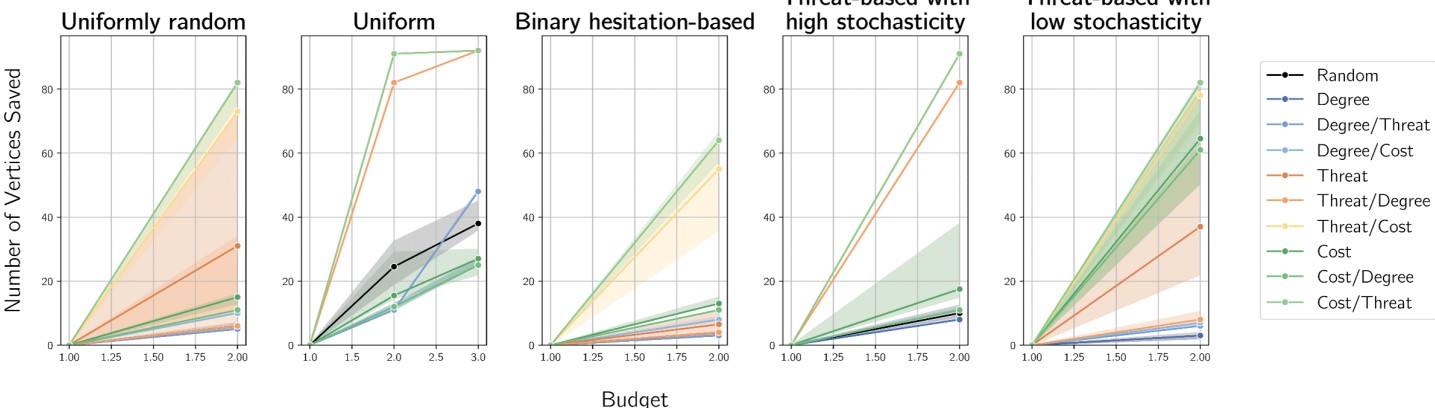

**Fig 16**. Plots of median numbers of vertices saved on caveman graphs on 100 vertices arranged in 10 cliques of size 10.

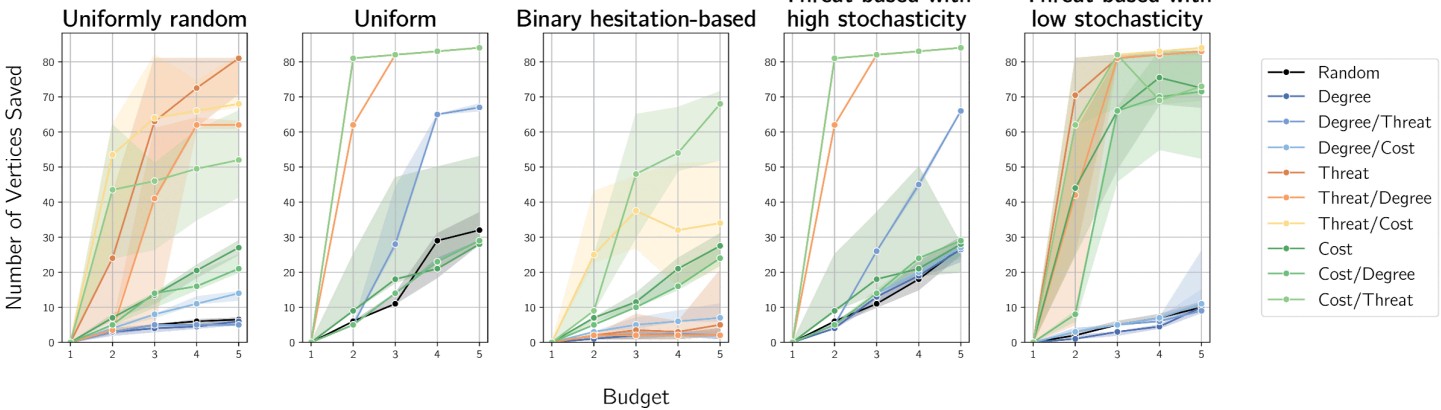

**Fig 17**. Plots of median numbers of vertices saved on caveman graphs on 100 vertices arranged in 20 cliques of size 5.

with maximum length of a path in the graph $\ell - 1$ in $\mathcal{O}((n+m)n^{b(\ell-2)})$ time (Theorem 7). This implies that Cost-Fire is in XP when parameterised by $\ell$ and the budget. Compare this to the result that Cost-Fire is FPT when parameterised by the sum of the treewidth of the input graph, the budget and the maximum lifetime of the problem, which we proved by expressing the problem in a fragment of second-order logic (Theorem 11).

## 7.1 Limitations

When interpreting the results of this study, several limitations should be kept in mind. First, the limited scale of graphs used (100 vertices for randomly generated graphs, 24-60 vertices for animal contact graphs) means that conclusions about heuristic performance can only be tentative, as our choice of graph classes may not fully capture real-world network heterogeneity. Also, none of our heuristics guarantee optimality, meaning comparisons are only relative rather than absolute solution quality assessments. Further, while our cost function examples are motivated by real-world observations

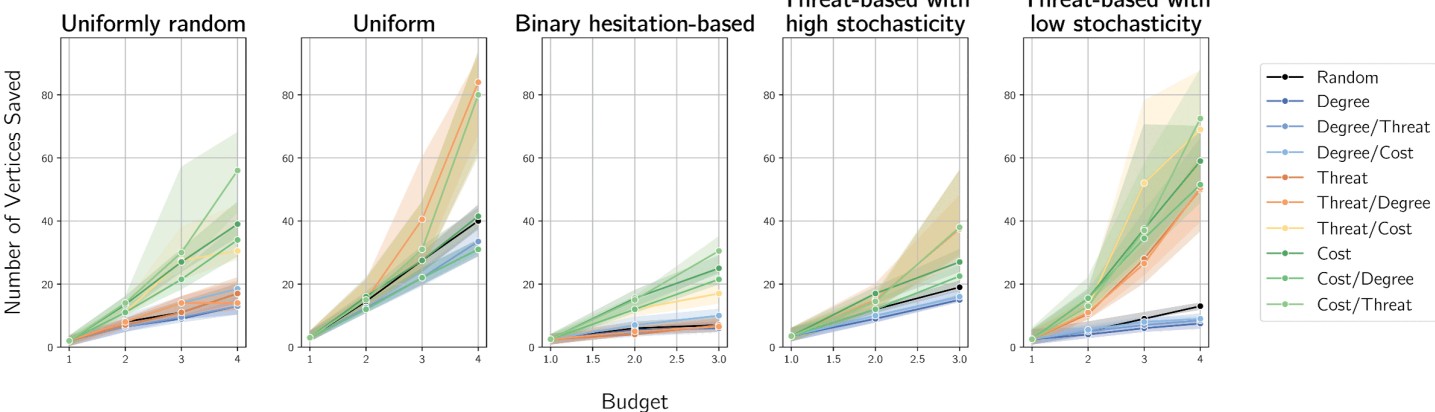

**Fig 18. Median numbers of vertices saved on randomly generated geometric graphs on 100 vertices with radius 0.15.**

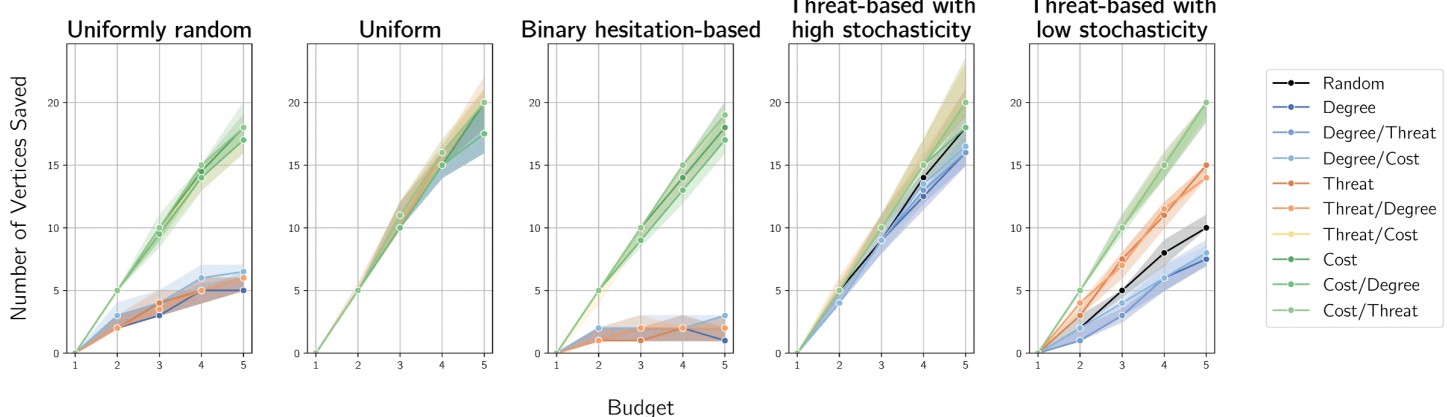

**Fig 19. Median numbers of vertices saved on randomly generated geometric graphs on 100 vertices with radius 0.25.**

(such as vaccine hesitancy), they are very simple and do not yet model the complex interplay of social, economic, geographic and other factors involved in such behaviours. Instead of making claims about the effectiveness of heuristics on different cost functions, our main aim in Sect 5 was to show how the theoretical model COST-FIRE may be used practically. However, we note that the exponential scaling of the state space (in Definition 3) in general limits practical application to small-to-moderately sized networks, and larger graphs may require approximate state representations or sampling-based approaches.

## 7.2 Future work

Several promising directions for future research may emerge from our work on COST-FIRE. For instance, future models could incorporate more nuanced cost functions that better reflect individual behaviours or environmental factors, which we hope would provide deeper insights into defence strategies. Future studies should also involve larger and more diverse

### Heuristic performance comparison for Random Regular graphs with degree 3

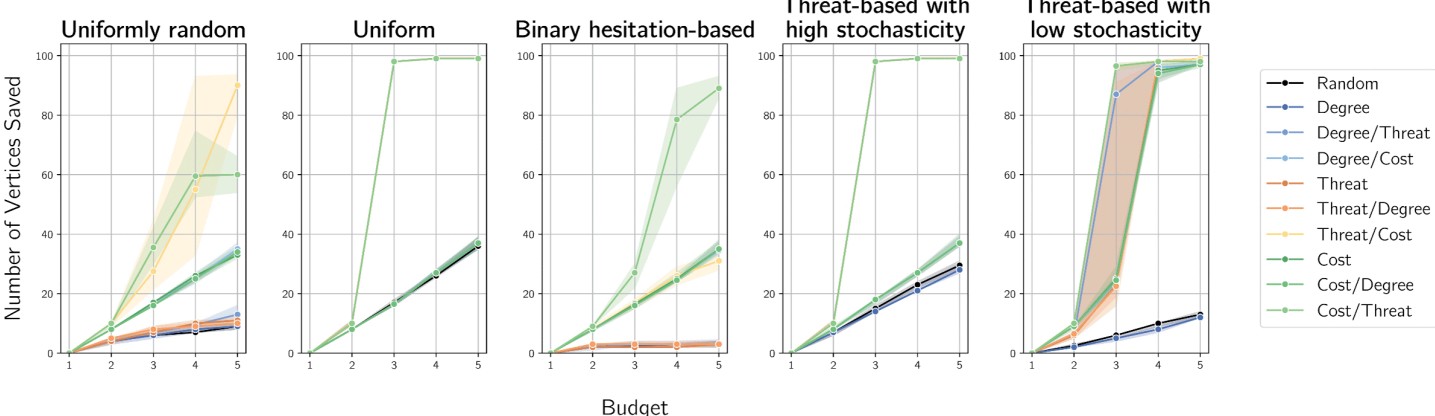

**Fig 20**. Plots of median numbers of vertices saved on randomly generated 3-regular graphs on 100 vertices.

### Heuristic performance comparison for Random Regular graphs with degree 8

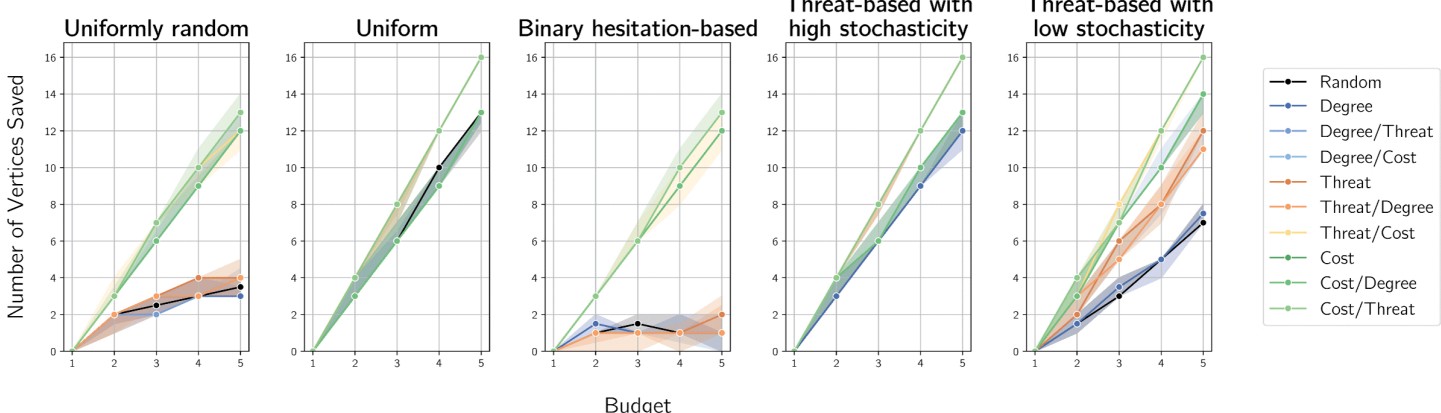

**Fig 21**. Plots of median numbers of vertices saved on randomly generated 8-regular graphs on 100 vertices.

real-world graphs, such as social networks, transportation systems and biological interaction networks, to evaluate scalability and practical applicability.

While we identify certain graph classes where Cost-Fire is tractable, a broader investigation into other graph structures is warranted. For example, exploring planar graphs, bipartite graphs, or graphs (not just trees) with bounded degree could reveal additional cases of computational feasibility. Understanding the boundaries of tractability across varied graph classes will enhance theoretical insights into the problem.

Although we have focused on unweighted and undirected graphs, Cost-Fire extends naturally to weighted and directed graphs. While complexity results in Sect 4 clearly continue to hold for weighted graphs, heuristics would need to be adapted and other heuristics not possible in the undirected and unweighted case may be favourable. In some directed graphs, we conjecture that the asymmetry for fire spread could be exploited to obtain stronger tractability results than possible on the underlying undirected graph, although in cases of strong connectivity, we expect similar hardness as in the undirected case.

On weighted or directed graphs, Cost-Fire remains valid without modification: costs can still depend on time, state, and vertex identity. The key differences lie in spreading and, consequently, in the reachability and time constraints that influence strategy design. Exploring these generalisations formally remains an open avenue for future work. We would be particularly interested in whether the fixed-parameter tractability result in Sect 4.5 extends under these richer dynamics.

To validate and extend our theoretical findings, we also recommend experiments on the graph classes used in the theoretical section using diverse cost functions and heuristic strategies. This would bridge the gap between theory and empirical performance, offering a more comprehensive understanding of the problem's behaviour across different contexts.

## Supporting information

**S1 Fig. Simulation result plots for Erdős-Rényi graphs on 100 vertices with probability 0.1.**
(TIFF)

**S2 Fig. Simulation result plots for Erdős-Rényi graphs on 100 vertices with probability 0.15.**
(TIFF)

**S3 Fig. Simulation result plots for Barabási-Albert graphs on 100 vertices with 1 edge per vertex.**
(TIFF)

**S4 Fig. Simulation result plots for Barabási-Albert graphs on 100 vertices with 3 edges per vertex.**
(TIFF)

**S5 Fig. Simulation result plots for Barabási-Albert graphs on 100 vertices with 5 edges per vertex.**
(TIFF)

**S6 Fig. Simulation result plots for clustered power-law graphs on 100 with 3 edges per vertex.**
(TIFF)

**S7 Fig. Simulation result plots for clustered power-law graphs on 100 with 4 edges per vertex.**
(TIFF)

**S8 Fig. Simulation result plots for Watts-Strogatz graphs on 100 vertices with mean degree 6.**
(TIFF)

**S9 Fig. Simulation result plots for Watts-Strogatz graphs on 100 vertices with mean degree 8.**
(TIFF)

**S10 Fig. Simulation result plots for randomly generated geometric graphs on 100 vertices with radius 0.2.**
(TIFF)

**S11 Fig. Simulation result plots for randomly generated 4-regular graphs on 100 vertices.**
(TIFF)

**S12 Fig. Simulation result plots for randomly generated 6-regular graphs on 100 vertices.**
(TIFF)

## Acknowledgments

This is a revised and extended version of a conference manuscript presented at the 13th International Conference on Complex Networks and their Applications (COMPLEX NETWORKS 2024) [37].

## Author contributions

**Conceptualization:** Ethan Hunter, Jessica Enright.

**Data curation:** Ethan Hunter.

**Formal analysis:** Ethan Hunter, Jessica Enright.

**Investigation:** Ethan Hunter.

**Methodology:** Ethan Hunter, Jessica Enright.

**Software:** Ethan Hunter.

**Supervision:** Jessica Enright.

**Validation:** Ethan Hunter, Jessica Enright.

**Visualization:** Ethan Hunter.

**Writing – original draft:** Ethan Hunter.

**Writing – review & editing:** Ethan Hunter, Jessica Enright.

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
