## [Decision Letter · Decision Letter 0]

24 Jul 2025

PONE-D-25-17298The Firefighter problem with dynamic defence costsPLOS ONE

Dear Dr. Hunter,

Thank you for submitting your manuscript to PLOS ONE. After careful consideration, we feel that it has merit but does not fully meet PLOS ONE’s publication criteria as it currently stands. Therefore, we invite you to submit a revised version of the manuscript that addresses the points raised during the review process.
**The three reviewers suggested accept, minor revision, and reject. I suggest major revision. I suggest addressing all the reviewers' comments. In addition, I think the graphs studied in the paper are very restricted. I suggest significantly improving the paper by considering more general graphs.**

We look forward to receiving your revised manuscript.

Kind regards,

Hoda Bidkhori

Academic Editor

PLOS ONE

**Journal Requirements:**

1. When submitting your revision, we need you to address these additional requirements.
Please ensure that your manuscript meets PLOS ONE's style requirements, including those for file naming. The PLOS ONE style templates can be found at https://journals.plos.org/plosone/s/file?id=wjVg/PLOSOne_formatting_sample_main_body.pdf and https://journals.plos.org/plosone/s/file?id=ba62/PLOSOne_formatting_sample_title_authors_affiliations.pdf
2. We noted in your submission details that a portion of your manuscript may have been presented or published elsewhere. ‘Yes, some results were published as a shorter (conference) manuscript, currently in print. Theoretical results were all briefly mentioned and justified in the original - they are fully proved here. Experimental results in the original were only preliminary and do not appear in this extended version. Instead, they have been replaced by more thorough simulation results and discussion”. Please clarify whether this [conference proceeding or publication] was peer-reviewed and formally published. If this work was previously peer-reviewed and published, in the cover letter please provide the reason that this work does not constitute dual publication and should be included in the current manuscript.
3. Thank you for uploading your study's underlying data set. Unfortunately, the repository you have noted in your Data Availability statement does not qualify as an acceptable data repository according to PLOS's standards.
At this time, please upload the minimal data set necessary to replicate your study's findings to a stable, public repository (such as figshare or Dryad) and provide us with the relevant URLs, DOIs, or accession numbers that may be used to access these data. For a list of recommended repositories and additional information on PLOS standards for data deposition, please see https://journals.plos.org/plosone/s/recommended-repositories.
4. We note that Figures 3 and 6 in your submission contain copyrighted images. All PLOS content is published under the Creative Commons Attribution License (CC BY 4.0), which means that the manuscript, images, and Supporting Information files will be freely available online, and any third party is permitted to access, download, copy, distribute, and use these materials in any way, even commercially, with proper attribution. For more information, see our copyright guidelines: http://journals.plos.org/plosone/s/licenses-and-copyright.
We require you to either present written permission from the copyright holder to publish these figures specifically under the CC BY 4.0 license, or remove the figures from your submission:
a. You may seek permission from the original copyright holder of Figures 3 and 6 to publish the content specifically under the CC BY 4.0 license. 
We recommend that you contact the original copyright holder with the Content Permission Form (http://journals.plos.org/plosone/s/file?id=7c09/content-permission-form.pdf) and the following text:“I request permission for the open-access journal PLOS ONE to publish XXX under the Creative Commons Attribution License (CCAL) CC BY 4.0 (http://creativecommons.org/licenses/by/4.0/). Please be aware that this license allows unrestricted use and distribution, even commercially, by third parties. Please reply and provide explicit written permission to publish XXX under a CC BY license and complete the attached form.”
Please upload the completed Content Permission Form or other proof of granted permissions as an "Other" file with your submission. 
In the figure caption of the copyrighted figure, please include the following text: “Reprinted from [ref] under a CC BY license, with permission from [name of publisher], original copyright [original copyright year].”
b. If you are unable to obtain permission from the original copyright holder to publish these figures under the CC BY 4.0 license or if the copyright holder’s requirements are incompatible with the CC BY 4.0 license, please either i) remove the figure or ii) supply a replacement figure that complies with the CC BY 4.0 license. Please check copyright information on all replacement figures and update the figure caption with source information. If applicable, please specify in the figure caption text when a figure is similar but not identical to the original image and is therefore for illustrative purposes only.
5. If the reviewer comments include a recommendation to cite specific previously published works, please review and evaluate these publications to determine whether they are relevant and should be cited. There is no requirement to cite these works unless the editor has indicated otherwise.

**Additional Editor Comments:**

The paper introduces a new variant of the classic Firefighter Problem, which models how contagions like rumors or diseases spread through a network. In the traditional problem, a player protects nodes in a graph to stop the spread and save as many as possible. This new version, called the Cost Function Firefighter Problem, adds complexity by making the cost of defending each node vary over time and depend on the graph's burning state. The authors prove that this version is computationally hard, even on tree structures where the original problem is more manageable. However, by formulating the problem in monadic second-order logic, they show it becomes fixed-parameter tractable when considering factors like treewidth, budget, and maximum time step.

In addition to theoretical analysis, the authors carry out empirical studies to compare different heuristic strategies—based on cost, threat, and degree—under varying cost conditions. They test these heuristics on both random and real-world graphs and find that the effectiveness of each approach depends heavily on the underlying graph structure. Notably, degree-based heuristics generally perform worse than strategies that consider the dynamic state of the graph, such as cost or threat.

The three reviewers suggested accept, minor revision, and reject. I suggest major revision. I suggest addressing all the reviewers' comments. In addition, I think the graphs studied in the paper are very restricted. I suggest significantly improving the paper by considering more general graphs.

Reviewers' comments:

Reviewer's Responses to Questions

**Comments to the Author**

1. Is the manuscript technically sound, and do the data support the conclusions?

Reviewer #1: No

Reviewer #2: Yes

Reviewer #3: Partly

2. Has the statistical analysis been performed appropriately and rigorously?

Reviewer #1: No

Reviewer #2: N/A

Reviewer #3: Yes

3. Have the authors made all data underlying the findings in their manuscript fully available?

Reviewer #1: Yes

Reviewer #2: Yes

Reviewer #3: Yes

4. Is the manuscript presented in an intelligible fashion and written in standard English?

Reviewer #1: No

Reviewer #2: Yes

Reviewer #3: Yes

5. Review Comments to the Author

**Reviewer #1:** Report for the manuscript is attached with the system metadata. It can be seen with the system login. I just suggest to prepare a manuscript that shows one track concept and results rather choosing multi directional concepts without technical concepts in the paper.

**Reviewer #2:** In the paper entitled "The Firefi ghter problem with dynamic defence costs",

the authors have introduced a variant of the Fire ghter problem called The Cost

Function Fire ghter Problem, in which each vertex has a cost to defend that varies

with time and the burning state of the graph. To complement theoretical fi ndings,

they compare performance of cost, threat and degree-based heuristics under various

cost functions. Their findings explain that the relative effectiveness of these heuris-

tics depends heavily on the structure of graph used in the process, but degree-based

heuristics tend to perform worse than state-based strategies. This manuscript has

been presented at an international conference, 2024.

I suggest to accept the manuscript, after authors address the following;

1. How well do these models capture real-world dynamics like delay, uncertainty,

or feedback loops?

2. Has the problem been simulated on real networks (e.g., social, transportation,

biological)?

3. Can the problem be extended to weighted or directed graphs, and how does

that affect solution strategies?

4. What algorithms are commonly used to solve or approximate solutions to the

Fire fighter Problem?

5. What are the best-known approximation ratios for specifi c versions of the

problem?

6. On which graph classes is the Fire fighter Problem solvable in polynomial time?

7. What are the known variants of the Firefi ghter Problem?

8. What are the limitations of this study?

9. Aligns text to both margins evenly.

10. Figure 8, 9, 10, 11 and 12 are not clear.

11. Make sure that all references are cited within the manuscript.

**Reviewer #3:** Manuscript ID: PONE-D-25-17298

Title: The Firefighter Problem with Dynamic Defence Costs

Strengths

• Excellent structure and clarity throughout.

• Strong theoretical foundation and clear NP-hardness reductions.

• Effective use of MSO logic for parameterized complexity.

• Experimental evaluation is thorough and includes both random and real-world graphs.

Minor Issues & Suggested Revisions

1. Abstract Improvements

• Line: “This problem is computationally hard, but can be solved efficiently...”

o Suggestion: Add a brief example or graph class for which it is efficiently solvable to make the sentence self-contained.

• Line: “We introduce a variant...”

o Suggestion: Consider emphasizing the novelty more clearly — e.g., highlight that it is the first to incorporate time-dependent vertex defence costs.

2. Notation and Definitions

• Definition 2 (Strategy)

o The multi-index notation (e.g., djid_{ji}) is clear, but it may help to include a small illustrative figure for visual clarity.

• Definition 3 (State Mapping)

o It would be beneficial to provide a visual timeline or state diagram for the example.

6. PLOS authors have the option to publish the peer review history of their article (what does this mean?). If published, this will include your full peer review and any attached files.

Reviewer #1: No

Reviewer #2: No

Reviewer #3: No

---

## [Author Response · Author response to Decision Letter 1]

5 Sep 2025

[Copied from the submitted file `Response to Reviewers.pdf` for convenience]

Response to reviews - PONE-D-25-17298

4th September 2025

Dear Dr. Bidkhori,

We thank you and the reviewers for your careful reading of our manuscript. We have revised the paper accordingly and write today to address each point in turn.

Journal Requirements

We have addressed the following reviews related to PLOS ONE’s specific submission requirements.

Journal Requirement 1 — Ensure that your manuscript meets PLOS ONE’s style requirements, including those for file naming.

Response: We apologise for the oversight. The figure and supplementary information files have all been named according to the style guides (FigX.abc for figures, SX_Fig.abc for supplementary figures).

Journal Requirement 2 — We noted in your submission details that a portion of your manuscript may have been presented or published elsewhere. Please clarify whether this [conference proceeding or publication] was peer-reviewed and formally published . . . please provide the reason that this work does not constitute dual publication and should be included in the current manuscript.

Response: This manuscript has been submitted by invitation to a special edition of the ‘Complex Networks and their Applications’ conference collection. The original conference article, presented at the Thirteenth International Conference on Complex Networks and

their Applications (COMPLEX NETWORKS 2024), was peer reviewed prior to publication. It is shorter than this manuscript (10 pages) and contains only sketches of proofs and limited preliminary experimental results as proof-of-concept. The current manuscript: adds to background and preliminary discussions to provide better context for the new problem, includes the proofs to new results, and presents the results of experiments proposed in the initial shorter paper. Hence, this manuscript contains significant extensions and additions to the previous publication.

Journal Requirement 3 — Thank you for uploading your study’s underlying data set. Unfortunately, the repository you have noted in your Data Availability statement does not qualify as an acceptable data repository according to PLOS’s standards. At this time, please upload the minimal data set necessary to replicate your study’s findings to a stable, public repository (such as figshare or Dryad) and provide us with the relevant URLs, DOIs, or

accession numbers that may be used to access these data.

Response: We apologise that link was not working at time of previous submission. While the DOI has now been successfully coined, and the link is working when tested, we have performed a wider range of experiments, the data of which (as used to produce plots) are all available at: https://doi.org/10.6084/m9.figshare.30053002.v1.

Journal Requirement 4 — We note that Figures 3 and 6 in your submission contain copyrighted images. All PLOS content is published under the Creative Commons Attribution License (CC BY 4.0), which means that the manuscript, images, and Supporting Information files will be freely available online, and any third party is permitted to access, download, copy, distribute, and use these materials in any way, even commercially, with proper attribution. We require you to either present written permission from the copyright holder to publish these figures specifically under the CC BY 4.0 license, or remove the figures from your submission

Response: We apologise for this oversight, a result of a misunderstanding of the CC BY-SA 3.0 licence under which the images were originally distributed. These figures, and references to them, have been removed in the updated manuscript.

Journal Requirement 5 — If the reviewer comments include a recommendation to cite specific previously published works, please review and evaluate these publications to determine whether they are relevant and should be cited. There is no requirement to cite these works unless the editor has indicated otherwise.

Response: Thank you, we have reviewed the recommended papers (Reviewer Com- ment 3.11). We found those that refer to the Firefighter problem are not closely related to the work in this manuscript.

Academic Editor Comments

Academic Editor Comment — I think the graphs studied in the paper are very restricted. I suggest significantly improving the paper by considering more general graphs.

Response: We are grateful for this important suggestion. To address it, we have re-run simulations on all existing random graph classes - on 100 vertices, rather than 50 - and included other random graph types. Specifically, the full list of graphs on which we run experiments is now:

• A contact graphs of 60 sleepy lizards,

• A contact graph of 24 raccoons,

• Erdős-Rényi random graphs, 100 vertices, generation probabilities 0.05, 0.10, 0.15, 0.20 and 0.25,

• Barabási-Albert graphs, 100 vertices, added edges per vertex: 1, 2, 3, 5 and 10, • Clustered power-law graphs, 100 vertices, added edges per vertex: 2, 3, 4 and 5, • Watts-Strogatz graphs, 100 vertices, mean degrees 4, 6, 8 and 10,

• Cavemen graphs, on 10 and 20 cliques (100 vertices in total),

• Random geometric graphs, with radii 0.15, 0.2 and 0.25, and • Random r-regular, 100 vertices for r =3, 4, 6 and 8.

We hope that this much broader scope of experimentation, coupled with explicit discussion of the limitations of this scope in the new Limitations section (§ 7.1, addresses this concern.

Reviewer 1

Reviewer Comment 1.1 — I suggest to prepare a manuscript that shows one track concept and results rather choosing multi directional concepts without technical concepts in the paper.

Response: We have reworked the central narrative of the manuscript, to make clear that our approach is to first show the problem is NP-hard to solve in general, which motivates a two-pronged approach in searching for tractable instances and comparing heuristic performance experimentally to understand the hardness of the problem in specific cases.

Reviewer Comment 1.2 — . . . neither the paper is scientifically so sound to be considered for the publication in Plos One . . . However it could be valuable for the specific journal. The authors can consider specialized journal for the publication of this article.

Response: We appreciate this comment and respectfully point out that this manuscript has been submitted by invitation to a special edition of the Complex Networks and their Applications conference collection, which we hope this reviewer deems a suitable venue.

Reviewer Comment 1.3 — The authors have mentioned Graph theory in the paper but mathematical tools of graph theory are not used in any results.

Response: To better expose the methods and logical structure, we have reorganised the manuscript to include a theoretical background section (§ 3), in which we give graph theoretic preliminaries (§ 3.1). In the Firefighter problem, we defend vertices in a graph, so this background is required. In a new theoretical methods section (§ 3.2), we explain the main techniques used, so it is clear that this work lies in algorithmic rather than structural graph theory.

Reviewer 2

Reviewer Comment 2.1 — How well do these models capture real-world dynamics like delay, uncertainty, or feedback loops?

Response: We discuss, in a new paragraph at the end of § 3.4, how cost functions can be defined to model the dynamics mentioned (delay, uncertainty and feedback loops).

Reviewer Comment 2.2 — Has the problem been simulated on real networks (e.g., social, transportation, biological)?

Response: While our empirical study does include real interaction graphs (the sleepy lizard and raccoon networks), these are very small and intended to demonstrate usage of the model rather than to find statistically significant conclusions. We state this explicitly in the new limitations section.

Reviewer Comment 2.3 — Can the problem be extended to weighted or directed graphs, and how does that affect solution strategies?

Response: Yes,andhardnessresultslargelyhold(althoughweconjecturethatthestructure of certain directed graphs may be leveraged for tractability on unweighted underlying graphs on which the problem remains hard). We explain our thoughts on this in § 7.2 and leave these extensions as future work.

Reviewer Comment 2.4 — What algorithms are commonly used to solve or approximate solutions to the Firefighter Problem?

Response: We have given an explanation of known approximation results in the new § 2.1.2 on Approximation of the original Firefighter problem. Of course, approximation is the best we can hope for in general, since the problem is NP-hard (Theorem 1, § 2.1).

Reviewer Comment 2.5 — What are the best-known approximation ratios for specific versions of the problem?

Response: We provide an overview of existing approximation results in the new § 2.1.2.

Reviewer Comment 2.6 — On which graph classes is the Firefighter Problem solvable

in polynomial time?

Response: We give some well-known graph classes on which the Firefighter problem is tractable in the new § 2.1.1, with citations.

Reviewer Comment 2.7 — What are the known variants of the Firefighter Problem? Response: We discuss Variants of the Firefighter problem in the new § 2.1.1, with

particular focus on variants related to the Cost Function variant we introduce.

Reviewer Comment 2.8 — What are the limitations of this study?

Response: We are particularly grateful for the reviewer pointing out the omission of a limitations section, which we have introduced as § 7.1.

Reviewer Comment 2.9 — Aligns text to both margins evenly.

Response: We have verified no ‘hbox-full’ warnings are produced on compilation in LATEX,

and have not amended or adjusted the formatting in the PLOS-ONE LATEX template.

Reviewer Comment 2.10 — Figures 8, 9, 10, 11 and 12 are not clear.

Response: We have re-generated all figures in higher resolution, as PDF files to avoid compression, and increased font sizes. We hope this improves the clarity of these figures.

Reviewer Comment 2.11 — Make sure that all references are cited within the manuscript. Response: We have manually checked and believe all references listed are cited at least

once in the main text of the manuscript.

Reviewer 3

Abstract improvements

Reviewer Comment 3.1 — Line: “This problem is computationally hard, but can be solved efficiently. . .” Suggestion: Add a brief example or graph class for which it is efficiently solvable to make the sentence self-contained.

Response: We have made this change in the abstract, both for the Firefighter problem (in the first paragraph of the abstract) and when introducing new variant (second paragraph).

Reviewer Comment 3.2 — Line: “We introduce a variant...” Suggestion: Consider emphasizing the novelty more clearly - e.g., highlight that it is the first to incorporate time-dependent vertex defence costs.

Response: Thankyou,wehavemadethischange(‘WedefineTheCostFunctionFirefighter Problem, the first variant of the Firefighter problem to introduce costs to defend each vertex that depend on time and problem state.’)

Notation and definitions

Reviewer Comment 3.3 — Definition 2 (Strategy) The multi-index notation (e.g., djidji [sic]) is clear, but it may help to include a small illustrative figure for visual clarity.

Response: This is a helpful suggestion - we have provided a figure (Figure 1, § 3.3) to illustrate a strategy for an instance of the Cost Function Firefighter problem that defends more than one vertex in the first turn, and added text (E.g. 1, after Definition 2) showing how this is represented using this notation.

Reviewer Comment 3.4 — Definition 3 (State Mapping) It would be beneficial to provide a visual timeline or state diagram for the example.

Response: This is also a helpful suggestion. We have provided (E.g. 2, after Definition 3) a fully worked example of the state mapping for the illustration used in our response to Reviewer Comment 3.3 for each turn in the lifetime of the instance.

Typographical & language refinements

Reviewer Comment 3.5 — Throughout: Minor grammar enhancements can improve flow: “Letthetargetk=n+3mk=n+3m,thebudgetb=2b=2andoverthesetof times...” Changeto: “Letthetargetbek=n+3mk=n+3m,withabudgetb=2b=2, over the time steps. . .”

Response: Thank you, we have made this change throughout and performed further proof reading with a particular focus on grammar and flow.

Reviewer Comment 3.6 — Page 2, Line 23: “We have designed Cost-Fire to model pop- ulation heterogeneity. . .” Suggestion: Replace “have designed” with “designed” for consistency in tense

Response: Respectfully, we believe the past perfect (rather than simple) is the correct tense here.

Reviewer Comment 3.7 — Page 6, Example 5: Consider formatting the piecewise function using LaTeX-style brackets or clearer indentation.

Response: This was originally formatted with a case environment, so to address this point we have increased the whitespace and hope this improves the readability of the function. We have also added more text explaining the examples that are more difficult to parse.

Table formatting

Reviewer Comment 3.8 — Table 1: Summary is excellent, but could use clearer column separation and improved caption wording: Caption Suggestion: “Summary of tractability and hardness results for Fire and Cost-Fire across graph classes.”

Response: We have increased the available spacing, moved the section numbers to the relevant cells and amended the caption in line with this suggestion.

Figures

Reviewer Comment 3.9 — Figures 1, 2, 3, 4, etc. Suggestion: Ensure all figures have consistent formatting and high resolution. Captions should explicitly state what is being shown and its relevance to the discussion.

Response: We have re-generated and saved all figures as higher-resolution files and checked all captions for clarity and completeness.

Section 3.6 – PathContainable

Reviewer Comment 3.10 — Excellent theory, but the term “PathContainable” could be better motivated. Consider including a brief paragraph before the definition to provide an intuitive explanation of the motivation and a real-world analogy (e.g., proximity-based decision-making in epidemics).

Response: Thank you for this suggestion, we have added this discussion to § 4.4 (second paragraph). Here, we explain the motivation for the PathContainable property, and how it relates to the path-containing strategies for the classic and budget Firefighter problems.

Cite the latest work

Reviewer Comment 3.11 — Cite the latest work.

Response: We have carefully reviewed the suggested papers and, while interesting, have

not found them sufficiently relevant to our work to cite them in the manuscript.

We thank you and the reviewers once again for your time and effort in considering this article.

Yours sincerely,

Ethan Hunter, on behalf of the authors.

---

## [Decision Letter · Decision Letter 1]

5 Jan 2026

The Firefighter problem with dynamic defence costs

PONE-D-25-17298R1

Dear Dr. Hunter,

We’re pleased to inform you that your manuscript has been judged scientifically suitable for publication and will be formally accepted for publication once it meets all outstanding technical requirements.

Kind regards,

Jan Rychtář

Academic Editor

PLOS One

Additional Editor Comments (optional):

Reviewers' comments:

Reviewer's Responses to Questions

**Comments to the Author**

1. If the authors have adequately addressed your comments raised in a previous round of review and you feel that this manuscript is now acceptable for publication, you may indicate that here to bypass the “Comments to the Author” section, enter your conflict of interest statement in the “Confidential to Editor” section, and submit your "Accept" recommendation.

Reviewer #1: (No Response)

Reviewer #2: All comments have been addressed

Reviewer #3: All comments have been addressed

2. Is the manuscript technically sound, and do the data support the conclusions?

Reviewer #1: No

Reviewer #2: Yes

Reviewer #3: Yes

3. Has the statistical analysis been performed appropriately and rigorously?

Reviewer #1: No

Reviewer #2: N/A

Reviewer #3: Yes

4. Have the authors made all data underlying the findings in their manuscript fully available?

Reviewer #1: Yes

Reviewer #2: Yes

Reviewer #3: Yes

5. Is the manuscript presented in an intelligible fashion and written in standard English?

Reviewer #1: No

Reviewer #2: Yes

Reviewer #3: Yes

6. Review Comments to the Author

Reviewer #1: This paper is more theoretical and less mathematical. The Term Graph Theory is used but Graph Theoretic concepts are not studied in the fashion it is required. Most of the results are based on Theoretical Concepts rather Mathematical. Firefighter Problem is a good topic to study, but it is interesting only if studied through mathematical ways. I, therefore, suggest the authors to make the work more mathematical and less theoretical by using mathematical tools in general and graphical tools in structural properties.

Reviewer #2: (No Response)

Reviewer #3: I have carefully reviewed the manuscript in accordance with the journal’s guidelines. The study is original, technically sound, and clearly presented. The methodology is appropriate, and the results are well supported by the analysis. I did not identify any concerns related to dual publication, plagiarism, data integrity, or research and publication ethics. The manuscript does not appear to have been submitted elsewhere, and all references are properly cited.

The authors have adequately addressed relevant literature and clearly stated the contributions of their work. Overall, the paper meets the ethical and academic standards of the journal, and I have no additional ethical concerns to report.

7. PLOS authors have the option to publish the peer review history of their article (what does this mean?). If published, this will include your full peer review and any attached files.

Reviewer #1: No

Reviewer #2: No

Reviewer #3: **Yes:** Sikander Ali

---

## [Editor Report · Acceptance letter]

PONE-D-25-17298R1

PLOS One

Dear Dr. Hunter,

I'm pleased to inform you that your manuscript has been deemed suitable for publication in PLOS One. Congratulations! Your manuscript is now being handed over to our production team.

Kind regards,

on behalf of

Dr. Jan Rychtář

Academic Editor

PLOS One